# LABEL-FREE MITIGATION OF SPURIOUS CORRELATIONS IN VLMS USING SPARSE AUTOENCODERS

**Bharat Chandra Yalavarthi, Nalini Ratha, Venu Govindaraju**
University at Buffalo, Buffalo, NY, USA
`byalavar@buffalo.edu, nratha@buffalo.edu, govind@buffalo.edu`

## ABSTRACT

Vision-Language Models (VLMs) have demonstrated impressive zero-shot capabilities across a wide range of tasks and domains. However, their performance is often compromised by learned spurious correlations, which can adversely affect downstream applications. Existing mitigation strategies typically depend on additional data, model retraining, labeled features or classes, domain-specific expertise, or external language models posing scalability and generalization challenges. In contrast, we introduce a fully interpretable, zero-shot method that requires no auxiliary data or external supervision named DIAL (Disentangle, Identify, And Label-free removal). Our approach begins by filtering the representations that might be disproportionately influenced by spurious features, using distributional analysis. We then apply a sparse autoencoder to disentangle the representations and identify the feature directions associated with spurious features. To mitigate their impact, we remove the subspace spanned by these spurious directions from the affected representations. Additionally, for cases where prior knowledge of spurious features in a dataset is unknown, we introduce DIAL+ which can detect and mitigate the spurious features. We validate our method through extensive experiments on widely used spurious correlation benchmarks. Results show that our approach consistently outperforms or matches existing baselines in terms of overall accuracy and worst-group performance, offering a scalable and interpretable solution to a persistent challenge in VLMs.

## 1 INTRODUCTION

Contrastive image-language models like CLIP have become foundational components in numerous applications, largely due to their remarkable zero-shot generalization capabilities Radford et al. (2021); Cherti et al. (2023). By training on web-scale data, they eliminate the need for task-specific labeled datasets, enabling efficient and scalable solutions for a wide range of downstream tasks and generative pipelines Lu et al. (2025); Zhu et al. (2025); Adila et al. (2024). However, despite strong aggregate performance, these vision-language models (VLMs) often fail on specific demographic or semantic groups, exhibiting performance far below the average Zhu et al. (2025); Chuang et al. (2023a); Yang et al. (2023). This vulnerability stems from their tendency to learn spurious correlations relying on non-causal features that are coincidentally prevalent in the training data rather than the causal task-relevant attributes Li et al. (2025). A commonly cited example in the literature is when models rely on imaging artifacts rather than causal disease features to make medical diagnoses Lu et al. (2025); Li et al. (2025). Figure 2 shows some examples of these spurious correlations visualized through a heatmap. As these spurious correlations may not hold in real-world test data, the model's reliability and zero-shot promise are fundamentally undermined, raising serious concerns about fairness and robustness Varma et al. (2024); Chuang et al. (2023b).

In recent times, a growing body of work has sought to mitigate the spurious correlations in VLMs. Many works like Chuang et al. (2023b); Trager et al. (2023); Lauscher et al. (2020) have focused on the textual modality for debiasing, but do not address biases encoded in the visual representations. Also, methods like Lauscher et al. (2020) require domain expertise or manual specification of debiasing textual prompts. Other prominent methods Yang et al. (2023); Zhang & Ré (2022);

---

Code available at: `https://github.com/byalavar/DIAL`

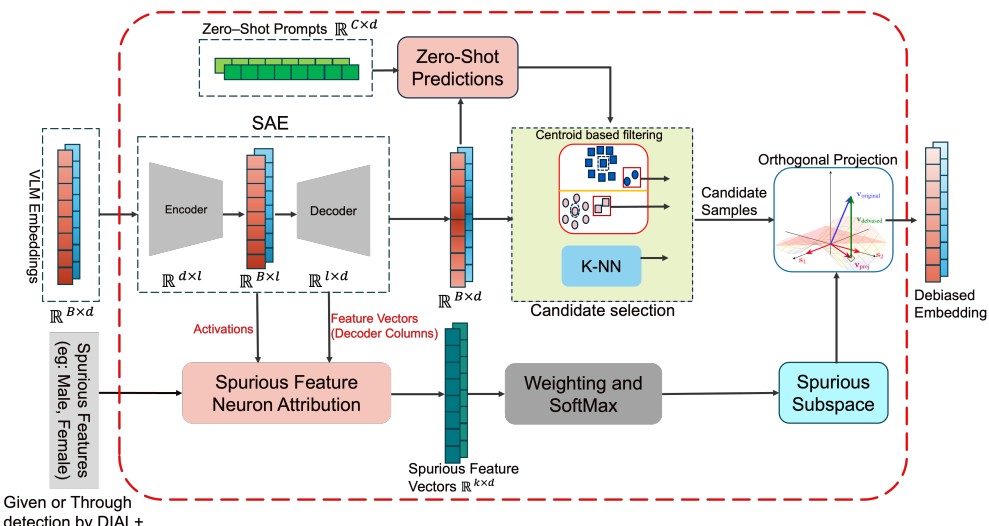

Figure 1: Overview of our proposed method. DIAL takes in VLM image embeddings and spurious features of a given dataset (e.g., "Male" and "Female" for the CelebA dataset). If DIAL+ is used, then there is no requirement for these spurious feature descriptions. The entire method operates in a zero-shot setting without requiring training, external data, class labels, or spurious feature labels.

Wang et al. (2023); Zhu et al. (2025) require fine-tuning the model or access to class and/or spurious feature labels, which negates the primary zero-shot advantage of VLMs. Recently, a few methods have emerged that operate in a truly zero-shot setting Lu et al. (2025); Adila et al. (2024); Chuang et al. (2023b). However, they introduce their own set of challenges. For instance, TIE Lu et al. (2025) relies on spurious feature labels for each sample to achieve optimal performance, which are often unavailable and expensive to acquire. Moreover, although it offers a label-free variant (TIE*), both implementations practically depend on additional data to compute their scaling factors. Concurrently, methods like ROBOSHOT Adila et al. (2024) rely on Large Language Models (LLMs) to generate task-specific insights, introducing concerns about reliability, hallucination, and sensitivity to the choice of LLM Lu et al. (2025).

To address the challenges of the current methods in mitigating spurious correlations, we propose an interpretable algorithm, DIAL (Disentangle, Identify, And Label-free removal), which works in a complete zero-shot setting without requiring training, additional data, or labels (both class labels and spurious feature labels). Our framework, when using DIAL requires two inputs: VLM embeddings of samples of a dataset and a high-level description of spurious features affecting the dataset (e.g., "Male", "Female" for CelebA). If DIAL+ is employed, it only requires VLM embeddings as it can detect the possible spurious features before mitigating them. Our mitigation method unfolds in three main steps. First, guided by the insight that samples affected by spurious features often deviate from their class centroids Li et al. (2025), we identify a candidate set of potentially biased samples without class labels using zero-shot predictions as pseudo-labels. Second, we employ Sparse Autoencoders (SAEs) to project these embeddings into a disentangled feature space. Within this space, we introduce a technique to reliably identify the feature directions that encode the spurious features. Finally, we debias the identified samples by removing the spurious subspace via an orthogonal projection. We also provide a technique to select the optimal parameters for our debiasing process, namely the fraction of features ($\alpha$) and the magnitude of subspace removal ($\lambda$). The overview of our proposed approach is given in Figure 1.

We conduct extensive experiments on five standard benchmark datasets, demonstrating the efficacy of our method compared to baselines. In summary, our contributions are:

- We propose **DIAL**, a fully zero-shot and interpretable framework designed to mitigate spurious correlations without requiring model training, additional data, class labels, or spurious feature annotations.

- To address scenarios where spurious attributes are unknown *a priori*, we introduce **DIAL+**, which autonomously detects and mitigates spurious correlations while maintaining performance comparable to DIAL.

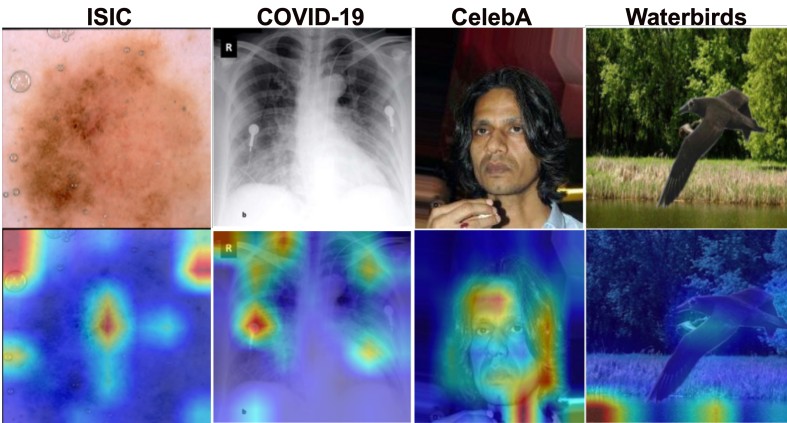

Figure 2: This figure illustrates how a CLIP model relies on spurious correlations for zero-shot predictions. For the ISIC dataset, it focuses on an image artifact instead of the lesion. For chest X-rays, it attends to a medical device rather than pneumonia indicators. On CelebA, it uses facial features instead of hair to identify "Blond hair", and for Waterbirds, it relies on the water background rather than the bird.

- We develop a novel technique to identify and isolate spurious feature subspaces directly from disentangled Sparse Autoencoder (SAE) representations in a zero-shot setting.
- We validate our approach across multiple benchmarks and VLM backbones, demonstrating that our method consistently outperforms or performs comparably to state-of-the-art baselines, while also establishing its efficacy in debiasing image retrieval.

## 2 RELATED WORK

**Mitigation with training or labels:** The problem of mitigating spurious correlations in deep learning models has been extensively studied. Techniques like Sagawa et al. (2019a); Liu et al. (2021); Yao et al. (2022); Krueger et al. (2021); Lu et al. (2024); Arjovsky et al. (2019); Idrissi et al. (2022); Yang et al. (2023); Goyal et al. (2023); Zhang & Ré (2022) aim to remove the effect of spurious correlations through reweighting the training samples, finetuning, regularization, or disparate loss functions. More recently Zhu et al. (2025) proposed to train a biased classifier to identify group labels and debias the classifier for VLMs. Li et al. (2025) identifies the minority samples using their dispersed distribution, and learns a transformation to a bias-invariant representation. Varma et al. (2024) shows that using region-level information in the images during training helps VLMs to ignore spurious correlations. All these methods require some form of training/fine-tuning, labels, or access to the model parameters. In contrast, our method works completely in a zero-shot setting without needing any labels, fine-tuning, or access to model parameters.

**Mitigation in zero-shot setting:** Several of the recent works on mitigating spurious correlations in VLMs focused on doing so in a zero-shot setting. Ge et al. (2023) proposes to augment text prompts with parent and child from WordNet hierarchy to improve zero-shot generalization. Trager et al. (2023) uses the average of text prompts, which are made from combining class labels with spurious features to get debiased text prompts for each class. Dehdashtian et al. (2024) uses reproducing kernel Hilbert spaces to debias CLIP's image and text representations. Chuang et al. (2023b) proposes a closed-form method through a calibrated projection matrix to remove biased direction from CLIP embeddings. Lu et al. (2025) mitigates spurious correlations by translating image embeddings along the direction of spurious vectors computed from text prompts. Its main algorithm needs access to spurious feature labels for each sample, so the authors also propose a variant that adapts when spurious feature labels are not present. Additionally, both variants of TIE require access to additional data to compute the scale parameter. Adila et al. (2024) uses LLMs to generate insights on spurious features, which are used to remove harmful components while keeping the useful ones. Unlike other zero-shot approaches, our method requires no auxiliary data for parameter tuning, no spurious feature labels, and no LLM for generating insights.

**Interpretable Methods for Mitigation:** Some of the works have proposed using interpretability methods for mitigating spurious correlations. Wu et al. (2023) proposes an iterative framework that discovers human-interpretable spurious concepts and intervenes on training data to mitigate their influence. Chakraborty et al. (2024) uses explainability-based heatmaps for creating pseudo labels to retrain and improve robustness to spurious features in an unsupervised manner. Karvonen et al. (2024) introduces a method to evaluate an SAE based on its capacity to mitigate spurious correlations. To do this, they train linear classifiers to identify specific neurons correlated with a known spurious attribute. The activations of these identified neurons are then ablated (i.e., zeroed out), and the resulting impact on model performance is measured. Unlike our approach, their method requires labeled training data and relies on activation zeroing rather than the removal of spurious subspace via orthogonal projection. Additionally, SAEs have been applied for concept erasure in diffusion models, Tian et al. (2025) finds unwanted concepts and deactivates them by modifying their activation with a temperature parameter. Recently, techniques to obtain contrastive sparse representations Wen et al. (2025) have been introduced, which could be used in combination with SAE for interpretability and mitigation applications.

# 3 METHODOLOGY

## 3.1 SETUP

Let $\mathcal{D} = \{(x_i, y_i)\}_{i=1}^{n}$ be a dataset with labels $y_i \in \mathcal{Y}$. A VLM uses an image encoder $\phi_v$ and a text encoder $\phi_t$ to map inputs into a $d$-dimensional embedding space $\mathbb{R}^d$.

For zero-shot classification, a set of class prompts (e.g., "a photo of $c$") are tokenized and then embedded by the text encoder to produce a set of class vectors $\{p_c\}_{c=1}^{|\mathcal{Y}|}$, where $p_c = \phi_t(\text{prompt}_c)$. The probability that an image $x_i$ belongs to class $c$ with temperature parameter $\tau$ is computed as:

$$P(y = c \mid x_i) = \text{softmax}_c \left( \frac{1}{\tau} \cdot \text{CosSim}(\phi_v(x_i), p_c) \right)$$

The set of groups is defined as $\mathcal{G} = \mathcal{Y} \times \mathcal{A}$, where $\mathcal{Y}$ is the set of class labels and $\mathcal{A}$ is the set of spurious attributes. We measure robustness of a VLM using three metrics: overall accuracy ($Acc_{avg}$), worst-group accuracy ($Acc_{wg}$), and the performance gap ($Acc_{gap}$), defined as:

$$Acc_{wg} = \min_{g \in \mathcal{G}} Acc_g, \quad Acc_{gap} = Acc_{avg} - Acc_{wg}$$

The goal of our zero-shot mitigation strategy is to improve both $Acc_{avg}$ and $Acc_{wg}$, and minimize $Acc_{gap}$, without requiring training or access to any labels.

## 3.2 FINDING SPURIOUS FEATURES

Our strategy is to use an SAE to disentangle the VLM embeddings $e_i$ and isolate feature directions corresponding to spurious attributes. An SAE decomposes an embedding into a sparse, linear combination of monosemantic features that are interpretable.

Given an embedding $e \in \mathbb{R}^d$, an SAE computes sparse feature activations $z \in \mathbb{R}^l$ and a reconstructed embedding $\hat{e} \in \mathbb{R}^d$:

$$z = \text{act}(W_{enc}e + b_{enc}) \qquad \hat{e} = W_{dec}z + b_{dec}$$

Here, $W_{enc} \in \mathbb{R}^{d \times l}$ is the encoder weight matrix, and the decoder matrix $W_{dec} \in \mathbb{R}^{l \times d}$ contains the $l$ disentangled feature vectors $\{f_j\}_{j=1}^{l}$ as its columns. We refer to this set of vectors as the feature dictionary, $\mathcal{F}$.

For each spurious attribute $a \in \mathcal{A}$ (e.g., "male" or "female"), we identify a subset of feature vectors $K_a \subset \mathcal{F}$ that strongly correlate with it. To do this, we adapt the attribution score method from Karvonen et al. (2024) to a zero-shot setting. First, we use the VLM's zero-shot classification ability to partition the reconstructed embeddings $\{\hat{e}_i\}$ from our dataset $\mathcal{D}$ into a positive set $P_a$ (samples exhibiting attribute $a$) and a negative set $N_a$. This is done using a prompt like "a photo of a $a$" and its negation.

The attribution score $S$ for each feature vector $f_j \in \mathcal{F}$ with respect to attribute $a$ is then calculated as:

$$S(f_j, a) = \left( \frac{1}{|P_a|} \sum_{i \in P_a} z_{i,j} - \frac{1}{|N_a|} \sum_{i \in N_a} z_{i,j} \right) \times \text{CosSim}(f_j, e_a)$$

where $z_{i,j}$ is the activation of feature $f_j$ for sample $i$, and $e_a = \phi_t(\text{prompt}_a)$ is the text embedding of the spurious attribute itself. This score is high when a feature's direction aligns with the attribute's semantic embedding and its activation is consistently higher for samples in the positive set.

Finally, to form the spurious feature set $K_a$, we select the top-$k$ features that account for a fraction $\alpha$ of the total attribution mass. We sort the features $f_j$ by $|S(f_j, a)|$ in descending order (indexed by $\pi$) and choose the smallest $k$ such that: $\sum_{j=1}^{k} |S(f_{\pi(j)}, a)| \geq \alpha \sum_{j=1}^{l} |S(f_j, a)|$

The resulting set $K_a = \{f_{\pi(1)}, \ldots, f_{\pi(k)}\}$ captures the primary directions in the embedding space associated with the spurious attribute $a$. The set $\mathcal{K} = \bigcup_{a \in \mathcal{A}} K_a$ contains all the feature vectors from every individual spurious feature set $K_a$

### 3.3 SPURIOUS FEATURE DETECTION:

To detect spurious features/concepts without relying on pre-defined attribute lists, we propose a data-driven detection method for DIAL+. This approach leverages the disentangled feature space of the SAE to isolate features that drive predictions in potentially biased samples.

**1. Identification of Influential Concepts.** First, we determine which disentangled concepts contribute decisively to the model's predictions. For a given sample $x_i$ with reconstructed embedding $\hat{e}_i$ and sparse activations $z_i$, we simulate the ablation of each feature $j$. Let $\hat{e}_{i,\neg j}$ denote the reconstruction obtained when the activation of feature $f_j$ is set to zero:

$$\hat{e}_{i,\neg j} = W_{dec}(z_i \odot (1 - \mathbf{1}_j)) + b_{dec}$$

where $\mathbf{1}_j$ is a one-hot vector at index $j$. We define the set of *influential concepts* $\mathcal{I}_i$ for sample $i$ as the set of features whose removal alters the zero-shot prediction of the sample:

We define the local influential concepts $\mathcal{I}_i$ and pool them to create a global set $\mathcal{I}_{pool}$ as follows:

$$\mathcal{I}_i = \left\{ j \in \{1, \ldots, l\} \mid \underset{c \in \mathcal{Y}}{\arg\max} \, P(c \mid \hat{e}_i) \neq \underset{c \in \mathcal{Y}}{\arg\max} \, P(c \mid \hat{e}_{i,\neg j}) \right\}, \quad \mathcal{I}_{pool} = \bigcup_{i=1}^{n} \mathcal{I}_i$$

**2. Candidate Sample Selection.** Next, we identify the subset of samples in the dataset that are likely affected by spurious correlations. We employ the Candidate Selection Algorithm (Alg. 1), which detects these samples (based on class centroid and k-NN inconsistency). Let $\mathcal{S}_{cand}$ denote the set of indices for the samples selected by the algorithm:

$$\mathcal{S}_{cand} = \{i \mid i \in \{1, \ldots, n\} \wedge \text{Algorithm 1}(e_i) \text{ returns True}\}$$

**3. Extraction of Spurious Concepts.** Finally, we identify the specific spurious features using the intersection of the pooled influential concepts ($\mathcal{I}_{pool}$) and the selected candidate samples ($\mathcal{S}_{cand}$). We compute the activation frequency $\nu_j$ for each feature $j \in \mathcal{I}_{pool}$ exclusively within the candidate set:

$$\nu_j = \sum_{i \in \mathcal{S}_{cand}} \mathbb{1}[j \in \mathcal{I}_i]$$

Features with high $\nu_j$ represent concepts from the influential pool that are consistently active in causing samples to deviate toward incorrect class centroids or k-NN inconsistency. We select the top-$k$ most commonly activated concepts based on $\nu$ to form the final set of spurious concepts $\mathcal{K}$.

### 3.4 MITIGATING SPURIOUS FEATURES

Given the identified set of spurious feature vectors $\mathcal{K}$, we aim to debias the reconstructed VLM embeddings $\hat{e}_i$ by removing their components that lie in the subspace spanned by these features. To account for noise in the feature selection process, we first refine the spurious subspace by weighting

each feature $f_j \in \mathcal{K}$ based on its alignment with the mean direction of the set. First, we compute the mean vector $m$ of the spurious features: $m = \frac{1}{|\mathcal{K}|} \sum_{f_j \in \mathcal{K}} f_j$

Next, we compute a vector of alignment scores $s \in \mathbb{R}^{|\mathcal{K}|}$, where each element $s_j$ corresponds to a feature $f_j$: $s_j = \beta \cdot \mathrm{CosSim}(f_j, m)$ A weight vector $w$ is then derived by applying the softmax function to these scores, where $\beta$ is a temperature hyperparameter controlling sharpness: $w = \mathrm{softmax}(s)$

To further denoise the set, we prune the features by setting weights that fall below a specified percentile to zero, yielding a filtered set of feature vectors $\mathcal{K}_f \subseteq \mathcal{K}$ with corresponding non-zero weights. We then form a matrix $V_w$ whose columns are the weighted feature vectors $\{w_j f_j \mid f_j \in \mathcal{K}_f\}$. We perform QR decomposition on this matrix, $V_w = QR$, to obtain an orthonormal basis $Q$ for the refined spurious subspace. The projection of $\hat{e}_i$ onto this subspace is given by $\hat{e_{i,\mathrm{proj}}} = QQ^T \hat{e}_i$.

The final, debiased embedding $\hat{e_{i,\mathrm{clean}}}$ is obtained by subtracting this projection from the original embedding, scaled by a mitigation factor $\lambda \in [0, 1]$: $\hat{e_{i,\mathrm{clean}}} = \hat{e}_i - \lambda \hat{e_{i,\mathrm{proj}}}$

This procedure removes information correlated with the identified spurious concepts while preserving other essential features of the original VLM embedding. We employ a targeted mitigation strategy, applying orthogonal projection to remove spurious features only from a subset of samples identified by our candidate selection algorithm (Alg. 1). This algorithm is designed to pinpoint samples that are likely to be affected by spurious correlations, which often lead to misclassifications. Operating in a label-free, zero-shot setting, our approach builds on the insight from prior work Li et al. (2025) that biased samples often lie far from their true class centroid. We approximate these class centroids by using the VLM's own zero-shot predictions as pseudo-labels. To enhance the robustness of this selection against noise and outliers, we further refine the candidate set using a standard k-Nearest Neighbors (k-NN) algorithm.

Our framework has three key parameters: the number of neighbors $k_*$ for k-NN, the attribution mass threshold $\alpha$, and the mitigation strength $\lambda$. To select these values effectively, we propose a grid-search-based algorithm (Alg. 2) that optimizes a zero-shot score reflecting the alignment between sample embeddings and the identified spurious features.

## 4 EXPERIMENTS

### 4.1 DATASETS

Following prior work by Lu et al. (2025) in zero-shot spurious correlation mitigation, we use the five established benchmarks for evaluating our method. CelebA Liu et al. (2015), Waterbirds Koh et al. (2021), FMOW Christie et al. (2018) and two medical datasets ISIC Codella et al. (2019), and COVID-19 Cohen et al. (2020). All datasets except FMOW have two classes and two associated spurious features, while FMOW has 62 classes with 5 spurious features. In accordance with the prior work Lu et al. (2025); Adila et al. (2024), we define groups as a combination of class label and spurious feature. For FMOW, we define a group based on the spurious feature following the procedure given in Wu et al. (2023). For zero-shot classification, we use the same text prompts used in our prior work and evaluate all the baselines with the same text prompts. For example, for the CelebA dataset, the zero-shot text prompts we use are "a photo of a celebrity with dark hair" and "a photo of a celebrity with blonde hair".

### 4.2 BASELINES

We evaluate our proposed method against existing zero-shot mitigation methods, including TIE Lu et al. (2025), ROBOSHOT Adila et al. (2024), Ideal Words Trager et al. (2023), Orth-Cali Chuang et al. (2023b), and Perception CLIP An et al. (2024). We also include the zero-shot and GroupPrompt zero-shot performance as the baselines. As established by prior works Sagawa et al. (2019b), we compare on worst group accuracy ($Acc_{wg}$ - WG), average accuracy ($Acc_{avg}$ - Acc), and gap between Acc and WG ($Acc_{gap}$ - Gap). In the results, we group the baselines into two groups, one with methods that require auxiliary information through either additional data, class/spurious feature labels, or LLM insights for mitigation. This group includes Perception CLIP An et al. (2024), ROBOSHOT

Table 1: CelebA: Comparison of our mitigation method with baselines in terms of zero-shot classification. Note that DIAL requires an a priori list of spurious features, whereas DIAL+ automatically detects and mitigates them. Best performance is bolded, and the second best is underlined.

| Method | Setting Requirements | | | CLIP ViT-B/32 | | | CLIP ViT-L/14 | | |
|---|---|---|---|---|---|---|---|---|---|
| | Additional Data | Class/Spurious Feature Labels | LLM | AVG (↑) | WG(↑) | Gap(↓) | AVG (↑) | WG(↑) | Gap(↓) |
| PerceptionCLIP | ✗ | ✗ | ✓ | 80.32 | 76.46 | 3.86 | 81.41 | 78.70 | 2.71 |
| ROBOSHOT | ✗ | ✗ | ✓ | 84.77 | 80.52 | 4.25 | 85.54 | 82.61 | 2.93 |
| TIE | ✓ | ✓ | ✗ | 85.11 | 82.63 | 2.48 | 86.17 | 84.60 | 1.57 |
| TIE∗ | ✓ | ✗ | ✗ | 85.11 | 82.63 | 2.48 | 86.17 | 84.60 | 1.57 |
| Zero-Shot | ✗ | ✗ | ✗ | 84.27 | 78.89 | 5.38 | 81.20 | 73.35 | 7.85 |
| GroupPrompt | ✗ | ✗ | ✗ | 80.38 | 74.90 | 5.48 | 77.86 | 68.94 | 8.92 |
| Ideal Words | ✗ | ✗ | ✗ | 80.96 | 78.12 | 2.84 | **89.15** | 76.67 | 12.48 |
| Orth-Cali | ✗ | ✗ | ✗ | 82.31 | 77.92 | 4.39 | 81.39 | 77.69 | 3.70 |
| **DIAL (Ours)** | ✗ | ✗ | ✗ | **85.54** | **83.47** | 2.17 | 86.87 | **85.24** | 1.63 |
| **DIAL+ (Ours)** | ✗ | ✗ | ✗ | 85.28 | 83.42 | **1.86** | 86.54 | 85.15 | **1.39** |

Table 2: Waterbirds: Comparison of our mitigation method with baselines in terms of zero-shot classification. Note that DIAL requires an a priori list of spurious features, whereas DIAL+ automatically detects and mitigates them. Best performance is in bold, and the second best is underlined.

| Method | Setting Requirements | | | CLIP ViT-B/32 | | | CLIP ViT-L/14 | | |
|---|---|---|---|---|---|---|---|---|---|
| | Additional Data | Class/Spurious Feature Labels | LLM | AVG (↑) | WG(↑) | Gap(↓) | AVG (↑) | WG(↑) | Gap(↓) |
| PerceptionCLIP | ✗ | ✗ | ✓ | 82.50 | 59.78 | 22.72 | 86.74 | 54.12 | 32.62 |
| ROBOSHOT | ✗ | ✗ | ✓ | 71.92 | 54.41 | 17.51 | 64.43 | 45.17 | 19.26 |
| TIE | ✓ | ✓ | ✗ | 79.82 | 71.35 | 8.47 | 84.12 | 78.82 | 5.30 |
| TIE∗ | ✓ | ✗ | ✗ | 76.91 | 61.24 | 15.67 | 78.98 | 61.60 | 17.38 |
| Zero-Shot | ✗ | ✗ | ✗ | 68.48 | 41.37 | 27.11 | 83.72 | 31.93 | 51.79 |
| GroupPrompt | ✗ | ✗ | ✗ | 66.79 | 43.46 | 23.33 | 56.12 | 10.44 | 45.68 |
| Ideal Words | ✗ | ✗ | ✗ | **79.20** | **60.28** | **18.92** | **87.67** | 64.17 | 23.50 |
| Orth-Cali | ✗ | ✗ | ✗ | 69.19 | 54.99 | 14.20 | 86.31 | 58.56 | 27.75 |
| **DIAL (Ours)** | ✗ | ✗ | ✗ | 71.88 | 52.82 | 19.06 | 82.6 | 68.69 | 13.91 |
| **DIAL+ (Ours)** | ✗ | ✗ | ✗ | 68.48 | 42.26 | 26.22 | 82.25 | **69.18** | **12.47** |

Adila et al. (2024), and TIE/TIE∗Lu et al. (2025). The other group, which does not require any of these, is our proposed method, along with standard zero-shot, GroupPrompt classification, Ideal Words Trager et al. (2023), and Orth-Cali Chuang et al. (2023b). For a fair comparison, we divide the baseline methods into these two groups in the results.

## 4.3 BACKBONE MODELS

Following prior work Adila et al. (2024); Lu et al. (2025), we examine CLIP ViT-B/32 (OpenAI), and ViT-L/14 (Laion-2B) Radford et al. (2021); Cherti et al. (2023) as backbones for Waterbirds and CelebA datasets. For the FMOW dataset, we use ViT-L/14 (Laion-2B) model. For medical datasets ISIC and COVID-19, we use BiomedCLIP Zhang et al. (2023b). For disentangling the representations, we use the pre-trained Matryoksha Sparse Autoencoders (MSAE) Zaigrajew et al. (2025) for all the backbone models used in the experiments. Any other SAE trained for VLMs can also be used instead of MSAE. We have evaluated our method with additional backbones including (ViT-H-14-quickgelu, EVA02-E-14-plus, ViT-SO400M-14-SigLIP-384) Cherti et al. (2023) whose results are presented in the appendix.

## 4.4 RESULTS

**CelebA and Waterbirds:**

Table 3: FMOW: Comparison of our mitigation method with baselines in terms of zero-shot classification. Note that DIAL requires an a priori list of spurious features, whereas DIAL+ automatically detects and mitigates them. Best performance is bolded, and the second best is underlined.

| Method | Setting Requirements | | | AVG (↑) | WG(↑) | Gap(↓) |
|---|---|---|---|---|---|---|
| | Additional Data | Class/Spurious Feature Labels | LLM | | | |
| PerceptionCLIP | ✗ | ✗ | ✓ | 17.70 | 12.61 | 5.09 |
| ROBOSHOT | ✗ | ✗ | ✓ | 19.79 | 10.88 | 8.91 |
| TIE | ✓ | ✓ | ✗ | 26.62 | 20.19 | 6.43 |
| TIE∗ | ✓ | ✗ | ✗ | 26.65 | 19.84 | 6.81 |
| Zero-Shot | ✗ | ✗ | ✗ | 26.02 | 18.06 | 7.96 |
| GroupPrompt | ✗ | ✗ | ✗ | 14.69 | 8.75 | **5.94** |
| Ideal Words | ✗ | ✗ | ✗ | 20.21 | 11.14 | 9.07 |
| Orth-Cali | ✗ | ✗ | ✗ | 26.11 | 19.45 | 6.66 |
| **DIAL (Ours)** | ✗ | ✗ | ✗ | 26.09 | **19.90** | 6.19 |
| **DIAL+ (Ours)** | ✗ | ✗ | ✗ | **26.23** | 19.24 | 6.99 |

Table 4: Medical Datasets - ISIC and COVID-19: Comparison of our mitigation method with baselines in terms of zero-shot classification. Note that DIAL requires an a priori list of spurious features, whereas DIAL+ automatically detects and mitigates them. Best performance is bolded, and the second best is underlined.

| Method | Setting Requirements | | | ISIC | | | COVID-19 | | |
|---|---|---|---|---|---|---|---|---|---|
| | Additional Data | Class/Spurious Feature Labels | LLM | AVG (↑) | WG(↑) | Gap(↓) | AVG (↑) | WG(↑) | Gap(↓) |
| PerceptionCLIP | ✗ | ✗ | ✓ | 52.74 | 41.55 | 11.19 | 56.87 | 48.84 | 8.03 |
| ROBOSHOT | ✗ | ✗ | ✓ | 59.84 | 53.30 | 6.54 | 53.10 | 32.75 | 20.35 |
| TIE | ✓ | ✓ | ✗ | 69.90 | 65.87 | 4.03 | 62.50 | 52.17 | 10.33 |
| TIE∗ | ✓ | ✗ | ✗ | 71.68 | 61.11 | 10.57 | 61.08 | 50.22 | 10.86 |
| Zero-Shot | ✗ | ✗ | ✗ | 70.21 | 42.21 | 28.00 | **61.81** | 44.83 | 16.98 |
| GroupPrompt | ✗ | ✗ | ✗ | 30.05 | 12.13 | 17.92 | 48.27 | 27.58 | 20.69 |
| Ideal Words | ✗ | ✗ | ✗ | 53.07 | 41.42 | 11.65 | 56.84 | 23.53 | 33.31 |
| Orth-Cali | ✗ | ✗ | ✗ | **72.54** | 21.43 | 51.11 | 51.72 | 44.83 | **6.89** |
| **DIAL (Ours)** | ✗ | ✗ | ✗ | 70.71 | **68.42** | **2.29** | 61.11 | **48.28** | 12.83 |
| **DIAL+ (Ours)** | ✗ | ✗ | ✗ | 68.93 | 65.45 | 3.48 | 61.11 | **48.28** | 12.83 |

On the CelebA dataset (results in Table 1), our method demonstrates superior performance, particularly with the ViT-B/32 backbone. It surpasses all zero-shot baselines across all three metrics, even outperforming methods that require auxiliary data, spurious feature labels, or the use of LLMs. When using the stronger ViT-L/14 backbone, our approach continues to achieve the highest worst group accuracy, lowest performance gap, underscoring its robust efficacy in mitigating spurious correlations.

For the Waterbirds dataset (results in Table 2), using the ViT-L/14 backbone, our method yields significant improvements in worst group accuracy and effectively reduces the performance gap compared to the baselines. We hypothesize that the performance on this dataset is influenced by the inherent complexity of the spurious attributes. The concepts of "land background" and "water background" are highly varied and complex, making it challenging to fully capture the corresponding feature space using only a high-level semantic description. This ambiguity may impact the precision of our attribution score calculation, explaining why some baselines perform better in certain configurations.

**FMOW:** We next evaluate our method on the challenging FMOW dataset (results in Table 3). Owing to the complicated nature of the dataset, following prior work Lu et al. (2025), we use only the ViT-L/14 backbone. Our method improves over the baselines in our sub-group on the worst group accuracy while still maintaining a comparable average accuracy.

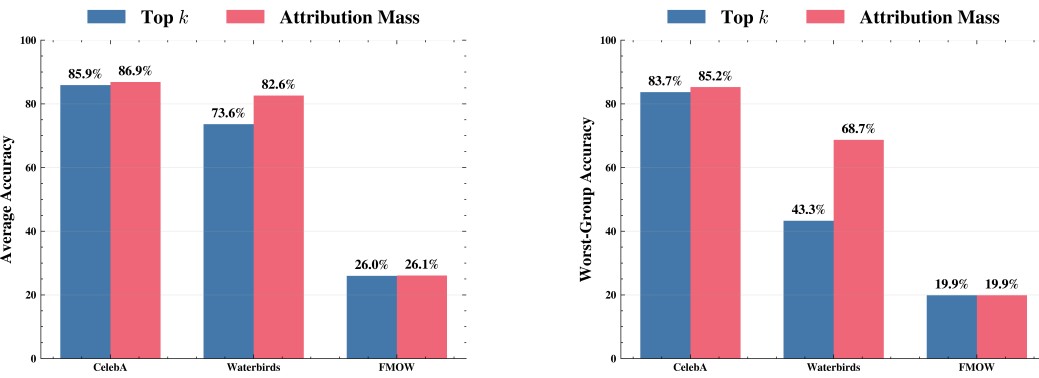

Figure 3: Comparison of spurious feature selection strategies.

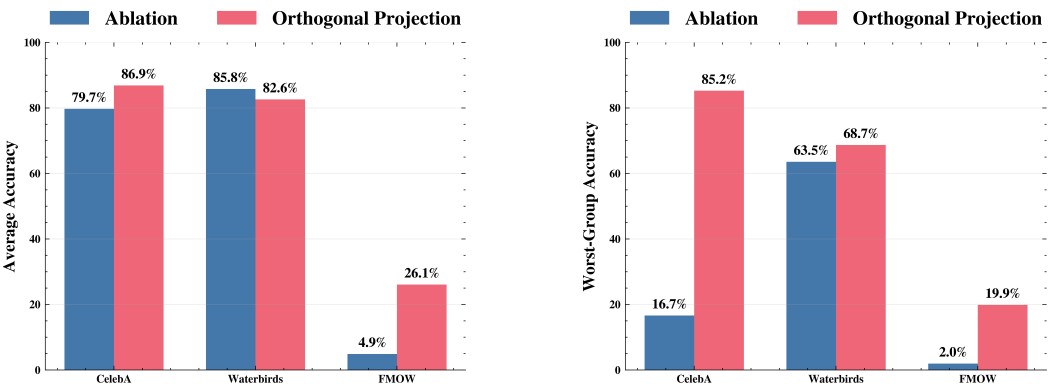

Figure 4: Comparison of spurious feature removal techniques.

**Medical Datasets:** The results on the medical datasets are presented in Table 4. On the **ISIC dataset**, our method demonstrates a substantial improvement in worst group accuracy and a corresponding reduction in the performance gap compared to all baselines. Notably, our fully zero-shot approach surpasses even those methods that rely on auxiliary data or additional labels for debiasing. Similarly, for the **COVID-19 dataset**, our approach improves over baselines in worst-group performance; it achieves this while maintaining a highly competitive average accuracy.

## 4.5 DEBIASED RETRIEVAL

Beyond zero-shot classification, we evaluate the efficacy of our method in debiasing image retrieval. Following the experimental protocol of Chuang et al. (2023b), we perform retrieval based on the cosine similarity between the query text and image embeddings from the FairFace benchmark Kärkkäinen & Joo (2019). To quantify fairness, we employ the MaxSkew@$k$ metric Geyik et al. (2019), which assesses the maximum logarithmic deviation between the observed frequency of a sensitive attribute in the top-$k$ results and a perfectly uniform distribution. We observe consistent reductions in MaxSkew scores across Age, Gender, and Ethnicity attributes compared to the original zero-shot baseline (ViT-L/14 trained on LAION-2B). These results (Table 5) demonstrate that DIAL effectively mitigates bias within the embedding space, resulting in fairer retrieval outcomes.

Table 5: Evaluation of our framework with image retrieval task on FairFace.

| Sensitive Feature | Original (MaxSkew@1000) (↓) | DIAL (MaxSkew@1000) (↓) |
|---|---|---|
| Age | 1.32 | **0.95** |
| Gender | 0.30 | **0.11** |
| Ethnicity | 0.61 | **0.32** |

### 4.6 ABLATIONS

In this section, we justify the technical choices made in our framework through a series of empirical studies. We focus on techniques to select the optimal spurious feature vectors and the removal of the spurious features. For the results reported in ablation studies, datasets CelebA, Waterbirds, and FMOW are used with ViT-L/14 as the backbone. Additional experiments concerning SAE selection, the relationship between SAE quality and performance, and further ablation studies are provided in the Appendix.

**Selection through top $k$ features vs attribution mass ($\alpha$):** We compare the difference between selecting the top $k$ spurious feature directions and selecting $\alpha$ fraction of the attribution mass. When we run the proposed parameter search algorithm to optimize $k$ vs $\alpha$, we see that the latter provides better results as shown in Figure 3. This could be due to the varying representation of different features in the SAE. For example, a specific concept like "color patch" might be represented with fewer feature vectors than "land background".

**Orthogonal projection vs neuron ablation:** Prior works have used both these techniques for concept removal. In our experiments (results shown in Figure 4), we find that orthogonal projection is much more effective at removing the spurious features than just ablating the corresponding activations to zero. This may be attributed to orthogonal projection removing the entire spurious subspace, while ablating a specific set of neurons to zero may still leave some unidentified spurious feature vectors watering down the mitigation. On the other hand, orthogonal projection can affect non-spurious features if they are very close to spurious features.

## 5 DISCUSSION

**Modality and Scope:** While this work focuses on mitigating spurious correlations in image embeddings, our method is modality-agnostic and can be applied to any VLM embedding. Mitigating with image modality distinguishes our approach from most zero-shot baselines that primarily target the textual modality.

**Parameterization:** The choice of a pre-trained SAE, backbone feature extractor, and dataset can influence the optimal parameters $(\alpha, \lambda)$ for mitigation, as different settings yield varying levels of feature disentanglement. However, our zero-shot parameter search addresses this dependency by automatically identifying the optimal configuration. The algorithm optimizes towards embedding equidistance to spurious concepts, thereby reducing bias (for DIAL) or minimizing spurious sample coverage (as determined by Alg. 1 for DIAL+). We note that because our framework operates in a strict zero-shot, data-free regime, the search process relies on the hyperparameters governing candidate selection and the specified parameter search ranges. Future work could explore analytical solutions to further improve the performance of our framework and reduce these dependencies.

**Interpretable Mitigation:** A key advantage of our method over prior work is its inherent transparency and interpretability. In high-stakes domains, this transparency is crucial for building trust and ensuring reliability. Our framework allows for a direct inspection of the mitigation process, providing a clear mechanism to diagnose the root causes of model failures and perform targeted debugging.

## 6 CONCLUSION

While VLMs possess remarkable zero-shot capabilities, they are often compromised by spurious correlations from web-scale data. We introduce a fully unsupervised, zero-shot method to mitigate these biases directly in the embedding space. Using a pre-trained SAE, we disentangle features and remove identified spurious directions via orthogonal projection on image embeddings. We further extend this to detect and mitigate correlations without prior knowledge of spurious features. Crucially, our approach requires no additional data, training, labels, or external LLMs, distinguishing it from prior work. By targeting image embeddings rather than text, we provide a distinct debiasing alternative. Zero-shot classification experiments across five datasets with multiple backbones, alongside image retrieval tasks, show our method matches or outperforms state-of-the-art techniques. Future work could explore applications in unlearning and fairness.

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

## A  REPRODUCIBILITY STATEMENT

The data pre-processing techniques we used for these experiments are the default CLIP preprocessing transforms based on the backbone architecture Lu et al. (2025) . All the results reported are on the test set of the datasets. MSAE models are trained with the settings mentioned in the GitHub repository of Zaigrajew et al. (2025). For non-medical settings, we used the datasets mentioned in the MSAE for training the SAEs. For medical, we trained SAE with the BiomedCLIP embeddings of PMC-15M train set Zhang et al. (2023a). While the specific parameters used for our framework are presented, they can be derived by implementing the proposed parameter search along with the candidate selection algorithm.

## B  LLM USAGE:

We used LLMs to polish the write-up after verifying its output content. We also used LLMs precisely for searching purposes to find the relevant related works by prompting for related works based on a specific topic.

## C  IMPLEMENTATION DETAILS

MSAEs which are trained for each backbone model and used across datasets. For non-medical datasets, off-the-shelf MSAEs Zaigrajew et al. (2025) which are trained using the CC3M Sharma et al. (2018) dataset are employed. For medical datasets, MSAEs are trained using PMC-15M Zhang et al. (2023b). In all cases, the SAEs were trained using Reverse Weighting (RW) mode with an expansion factor of 8 for 30 epochs. The optimal hyperparameters for spurious feature selection ($\alpha$), mitigation strength ($\lambda$), k-NN neighborhood size ($k_*$), and text centroid weight ($w$) were determined using our parameter search algorithm with specific search ranges that vary based on the dataset and backbone capacity. The optimal parameter configurations utilized for DIAL to achieve our reported results are summarized in Table 6.

Table 6: DIAL hyperparameter configurations used across datasets and backbones for candidate selection and mitigation.

| Dataset | Backbone | $\alpha$ | $\lambda$ | $k_*$ | Text Weight ($w$) |
|---|---|---|---|---|---|
| CelebA | ViT-B/32 | 0.3 | 0.5 | 10 | 1 |
|  | ViT-L/14 | 0.3 | 0.6 | 10 | 1 |
| Waterbirds | ViT-B/32 | 0.75 | 0.9 | 10 | 0 |
|  | ViT-L/14 | 0.65 | 0.9 | 10 | 0 |
| FMoW | ViT-L/14 | 0.7 | 0.6 | 25 | 1 |
| COVID-19 | BiomedCLIP | 0.6 | 0.4 | 10 | 0 |
| ISIC | BiomedCLIP | 0.1 | 0.2 | 10 | 0 |

## D  ABLATIONS AND ANALYSIS:

We perform additional ablation studies to justify the design choices made in our framework and explain its effectiveness.

**Candidate Selection:**  We apply our subspace removal only on candidates that could be disproportionately affected by spurious correlations. We find this selection to improve the overall performance. Table 7 compares our selective candidate selection approach against a global application of the subspace removal method.

Table 7: Ablation study with and without candidate selection

| Dataset - Model | Without candidate selection | With Candidate selection |
|---|---|---|
| CelebA - ViT L/14 | 86.43/84.82 | **86.54/85.15** |
| CelebA - ViT B/32 | 84.85/81.67 | **85.28/83.42** |
| Waterbirds - ViT L/14 | 81.60/51.56 | **82.25/69.18** |
| Waterbirds - ViT B/32 | 70.45/50.31 | **71.88/52.82** |
| FMOW - ViT L/14 | 26.04/19.55 | **26.09/19.90** |

Table 8: Ablation study for identifying SAE features.

| Model - Dataset | Mean Activation Diff | Cosine Similarity | Both |
|---|---|---|---|
| CelebA - ViT L/14 | 86.86/**85.39** | 86.78/85.35 | **86.87**/85.24 |
| CelebA - ViT B/32 | 85.54/83.33 | 85.28/82.78 | **85.54/83.47** |
| Waterbirds - ViT L/14 | **82.61/68.85** | 74.95/50.29 | 82.6/68.69 |
| Waterbirds - ViT B/32 | 71.85/52.68 | 70.69/48.12 | **71.88/52.82** |
| ISIC | **70.71/68.42** | 62.45/55.84 | **70.71/68.42** |
| COVID-19 | **61.11/48.28** | 58.33/34.48 | **61.11/48.28** |

**Feature Attribution:** Motivated by prior work on identifying SAE features Karvonen et al. (2024), we used both the mean activation difference of positive and negative sample activations and cosine similarity of the SAE feature with the text embedding. The ablation experiment (Table 8) suggests that mean activation difference alone yields performance comparable to using both.

**Analysis:** In the four points below, we analyze the possible reasons for the efficacy of our framework.

**1) Minimized Feature Interference via Disentanglement:** Standard baselines often operate in dense, polysemantic embedding spaces. In such spaces, removing a spurious feature vector frequently degrades causal features due to feature superposition. By using the SAE latent space, we leverage a highly disentangled representation where feature vectors are nearly orthogonal. This orthogonality allows us to surgically remove spurious features with minimal impact on the semantic integrity of causal features.

**2) Selective Intervention using Candidate Selection:** Even within the SAE latent space, perfect orthogonality is not always achieved. Blanket removal of features across all samples can inadvertently harm "clean" samples (those not relying on spurious correlations). We explicitly identify samples that are disproportionately affected by spurious features (high activation of spurious components relative to causal ones). We apply our removal intervention only to these identified samples. This preserves the integrity of samples that are already robust, preventing the performance degradation seen when applying global intervention (Table 7).

**3) Subspace removal instead of feature ablation:** Removing the entire spurious subspace, rather than simply ablating the corresponding feature activation to zero, aids in removing unidentified spurious features that are highly aligned with that subspace. This results in a more effective elimination of spurious features, as demonstrated in Figure 4 of the paper.

**4) Feature selection through Attribution Mass:** Since different backbone embeddings and SAEs exhibit varying activation patterns, using a fixed Top-K approach to select feature directions corresponding to spurious features was shown to be less effective than selection through attribution mass, as shown in Figure 3 of the paper.

# E    ANALYSIS ON SAE ARCHITECTURE AND DATA

In this section, we evaluate the robustness of our framework across three dimensions: i) different SAE architectures (BatchTopKSAE Bussmann et al. (2024), JumpReLU Rajamanoharan et al. (2024)), ii) SAE quality metrics (reconstruction loss, sparsity, and decoder orthogonality), and iii) the impact of training SAEs on domain-specific debiasing datasets.

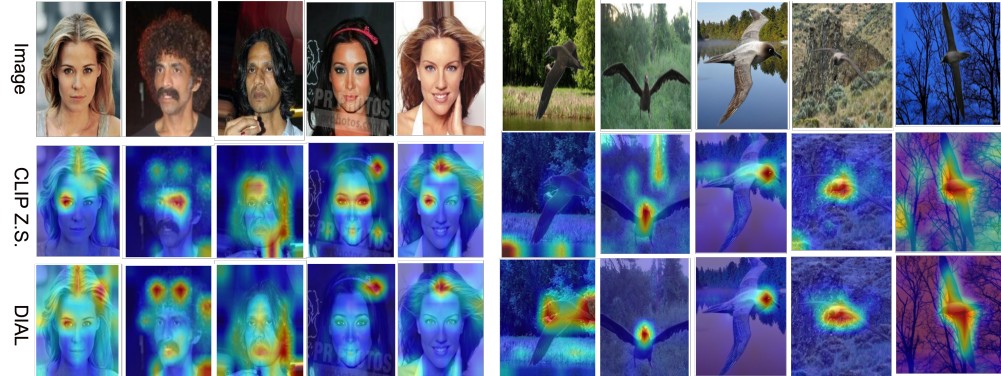

Figure 5: Visualization of the DIAL shifting model focuses from spurious to causal features. In the CelebA hair color classification task, DIAL redirects attention from gender-correlated attributes toward relevant regions (hair, eyebrows, and facial hair). Similarly, in the Waterbirds dataset, the method shifts focus from the background environment to the bird itself.

### E.1    IMPACT OF SAE ARCHITECTURE

Table 9 presents the performance of our framework when employing different SAE variants. We observe that MSAE achieves the best overall performance on average. This superiority is likely attributable to the quality of MSAE as measured by various evaluation metrics Zaigrajew et al. (2025).

Table 9: Evaluation of our framework using different SAEs. Average Accuracy/Worst-Group Accuracy are reported.

| Dataset | BatchTopKSAE | JumpReLUSAE | MSAE |
|---------|--------------|-------------|------|
| CelebA | 87.35/84.44 | **87.74**/84.44 | 86.87/**85.24** |
| Waterbirds | 71.98/41.51 | 77.80/56.01 | **82.6/68.69** |
| ISIC | 69.57/58.73 | **72.16**/66.67 | 70.71/**68.42** |
| COVID-19 | 46.52/20.69 | 58.33/34.48 | **61.11/48.28** |

### E.2    SAE QUALITY VS. MITIGATION PERFORMANCE

We analyze how intrinsic SAE properties, specifically reconstruction quality, sparsity, and decoder orthogonality, affect debiasing performance (Tables 10 and 11). Our analysis yields three key observations:

1. **Disentanglement Robustness:** High disentanglement capability (measured here via decoder orthogonality) persists in the evaluated SAEs even when reconstruction loss is relatively high.

2. **Dataset Sensitivity:** While our framework remains effective across varying sparsity levels, the Waterbirds dataset is more sensitive to reconstruction degradation than CelebA.

3. **General Improvement:** Worst Group (WG) accuracy improves across all evaluated configurations.

From these results, we infer that as long as an SAE reconstructs well enough to preserve discriminative features while maintaining high disentanglement, it can be effectively used to mitigate spurious correlations.

Table 10: Analysis of performance of our method w.r.t SAE Quality on CelebA dataset

| Recon. Loss ($\downarrow$) | Sparsity ($\uparrow$) | Decoder Orthogonality ($\downarrow$) | Avg/WG Acc ($\uparrow$) |
|---|---|---|---|
| 0.44 | 0.72 | 0.0016 | **87.35**/84.44 |
| 0.012 | 0.51 | 0.0014 | 86.41/**84.71** |
| 0.019 | 0.79 | 0.0016 | 86.48/84.65 |

Table 11: Analysis of performance of our method w.r.t SAE Quality on Waterbirds dataset

| Recon. Loss ($\downarrow$) | Sparsity ($\uparrow$) | Decoder Orthogonality ($\downarrow$) | Avg/WG Acc ($\uparrow$) |
|---|---|---|---|
| 0.44 | 0.72 | 0.0016 | 71.98/41.51 |
| 0.075 | 0.49 | 0.0013 | 75.2/49.31 |
| 0.019 | 0.79 | 0.0016 | **81.98/68.51** |

### E.3 Training on Debiasing Dataset

To assess the feasibility when large-scale pre-trained SAEs are unavailable, we trained SAEs from scratch on the specific debiasing datasets: CelebA ($\sim$140k images) and Waterbirds ($\sim$4.6k images). Table 12 shows that these domain-specific SAEs successfully improve worst-group accuracy. The performance on CelebA is comparable to that of the pre-trained SAE. However, the Waterbirds-trained SAE performs slightly worse, likely due to the limited training data size. These results demonstrate that training an SAE on the target dataset could be a feasible strategy for spurious correlation mitigation, particularly when data volume is sufficient.

Table 12: Evaluation of our framework when SAE is trained with debiasing dataset

| Dataset | Original (Avg/WG) ($\uparrow$) | DIAL (Avg/WG) ($\uparrow$) |
|---|---|---|
| CelebA | 81.20/73.35 | **86.41/84.71** |
| Waterbirds | **83.72**/31.93 | 75.2/**49.31** |
| FMOW | 26.05/18.16 | **26.67/18.16** |

## F Additional Backbones:

We have evaluated our framework on additional backbones whose results are provided in Tables 13, 14, 15.

Table 13: Evaluation of our framework with SigLIP (ViT-SO400M-14-SigLIP-384)

| Dataset | Original (Avg/WG) ($\uparrow$) | DIAL (Avg/WG) ($\uparrow$) |
|---|---|---|
| CelebA | 82.51/79.11 | **84.32/82.02** |
| Waterbirds | **80.92**/61.37 | 80.54/**66.16** |
| FMOW | **34.53**/25.18 | 34.10/**25.60** |

Table 14: Evaluation of our framework with EVA02-E-14-plus

| Dataset | Original (Avg/WG) ($\uparrow$) | DIAL (Avg/WG) ($\uparrow$) |
|---|---|---|
| CelebA | 84.78/80.54 | **87.76/86.52** |
| Waterbirds | **76.95**/37.85 | 76.78/**56.65** |
| FMOW | **29.62**/15.97 | 29.30/**16.70** |

Table 15: Evaluation of our framework with ViT-H-14-quickgelu

| Dataset | Original (Avg/WG) (↑) | DIAL (Avg/WG) (↑) |
|---|---|---|
| CelebA | 83.80/80.00 | **83.81/81.59** |
| Waterbirds | 85.60/51.09 | **88.47/63.86** |
| FMOW | 30.21/19.44 | **30.57/19.90** |

## G  INTERPRETABILITY EVALUATION

To evaluate the interpretability of our framework, we utilize the Monosemanticity Score (MS) proposed by Pach et al. (2025). We compute the MS score on the validation sets of the SAEs. Specifically, we use the ImageNet validation set for the standard vision backbones (ViT-B/32, ViT-L/14) and the PMC-15M validation set for the medical backbone (BiomedCLIP). This ensures that the interpretability scores reflect the general quality and complexity of the features learned by the SAE, independent of the specific downstream tasks.

We computed the MS scores using the automated method from Pach et al. (2025) across all neurons in our trained SAEs. The average per-neuron MS scores are reported in Table 16.

Table 16: Average Monosemanticity Scores (MS) for the SAEs used in our framework.

| SAE - Backbone Model | MS Score (Avg) |
|---|---|
| CLIP ViT-B/32 | 0.56 |
| CLIP ViT-L/14 | 0.43 |
| BiomedCLIP | 0.42 |

To contextualize these results, we refer to the user study conducted by Pach et al. (2025), which calibrates MS scores against human perception. Their study establishes that an MS score in the range of 0.4–0.5 corresponds to a human alignment rate of approximately 65–70%. Our results (0.42–0.56) indicate that the features learned by the SAEs are largely monosemantic and align with human perception. Consequently, this semantic coherence enables a verifiable explainability pipeline because the underlying units represent interpretable concepts, and human experts can explicitly interpret the spurious feature directions identified by our model and inspect the mitigation process.

## H  ALGORITHMS

---

**Algorithm 1** Candidate Selection

---

**Require:**
1: $E = \{e_i\}_{i=1}^n$: set of image embeddings.
2: $\hat{Y} = \{\hat{y}_i\}_{i=1}^n$: set of pseudo-labels from zero-shot predictions.
3: $T = \{c \to t_c\}$: map of class labels to text embeddings.
4: $k$: number of neighbors for k-NN.
5: $w$: text embedding weight.
  ▷ Calculate hybrid centroids for each class $c$
6: **for** each class $c \in \text{unique}(\hat{Y})$ **do**
7:     $\mu_c \leftarrow (1 - w) \cdot \text{Mean}(\{e_i \mid \hat{y}_i = c\}) + w \cdot T[c]$
8: **end for**
  ▷ Identify candidates based on centroid similarity or k-NN disagreements
9: $M_{centroid} \leftarrow [\text{argmax}_{c'} \text{CosSim}(e_i, \mu_{c'}) \neq \hat{y}_i]_{i=1}^n$
10: $M_{knn} \leftarrow \left[ \text{k-NN}(e_i, E, \hat{Y}, k) \neq \hat{y}_i \right]_{i=1}^n$
11: $M \leftarrow M_{centroid} \vee M_{knn}$  ▷ Combine candidate sets
12: **return** $M$

---

---

**Algorithm 2** Optimal Debiasing Parameter Search

---

**Require:** $E = \{e_i\}_{i=1}^n$, the set of original VLM embeddings. $t_{c1}, t_{c2}$, text embeddings for opposing spurious concepts (e.g., "male" and "female"). $S_k, S_w, S_\alpha, S_\lambda$, search ranges for parameters $k_*, w, \alpha,$ and $\lambda$.
**Ensure:** $\hat{k_*}, \hat{w}, \hat{\alpha}, \hat{\lambda}$, the optimal framework hyperparameters.
1: $score_{best} \leftarrow \infty$
2: **for** each $k \in S_k$ and $w \in S_w$ **do**
3:     $M \leftarrow \text{CANDIDATESELECTION}(E, k, w)$                          ▷ Algorithm 1
4:     $E_{sub} \leftarrow E[M]$                          ▷ Apply mask to get the candidate subset
5:     **if** $|E_{sub}| = 0$ **then continue**
6:     **end if**
7:     **for** each $\alpha \in S_\alpha$ **do**
8:         $Q \leftarrow \text{IDENTIFYSPURIOUSSUBSPACE}(E_{sub}, \alpha)$
9:         **if** $Q$ is not valid **then continue**
10:         **end if**
11:                          ▷ Search for the best global $\lambda$ for this mask and subspace
12:         **for** each $\lambda \in S_\lambda$ **do**
13:             $score_{current\_pair} \leftarrow 0$
14:             **for** each sample $e_i \in E_{sub}$ **do**
15:                 $e_{i,\text{clean}} \leftarrow e_i - \lambda(QQ^T e_i)$                          ▷ Apply orthogonal projection
16:                 $e_{i,\text{clean}} \leftarrow \text{Normalize}(e_{i,\text{clean}})$                          ▷ $L_2$ normalization
17:                                  ▷ Score measures neutrality between the two opposing concepts
18:                 $d_{current} \leftarrow |\text{CosSim}(e_{i,\text{clean}}, t_{c1}) - \text{CosSim}(e_{i,\text{clean}}, t_{c2})|$
19:                 $score_{current\_pair} \leftarrow score_{current\_pair} + d_{current}$
20:             **end for**
21:                                  ▷ Calculate the mean score for this parameter combination
22:             $score_{current\_pair} \leftarrow score_{current\_pair}/|E_{sub}|$
23:             **if** $score_{current\_pair} < score_{best}$ **then**
24:                 $score_{best} \leftarrow score_{current\_pair}$
25:                 $\hat{k_*}, \hat{w} \leftarrow k, w$
26:                 $\hat{\alpha}, \hat{\lambda} \leftarrow \alpha, \lambda$
27:             **end if**
28:         **end for**
29:     **end for**
30: **end for**
31: **return** $\hat{k_*}, \hat{w}, \hat{\alpha}, \hat{\lambda}$

---

