# OpenReview forum: "Label-Free Mitigation of Spurious Correlations in VLMs using Sparse Autoencoders"
_ICLR.cc/2026/Conference — ICLR 2026 Poster_

### Official Review · Reviewer_TSFp · 2025-10-28

**Soundness:** 3
**Presentation:** 3
**Contribution:** 3
**Rating:** 4
**Confidence:** 5

**Summary:**

This paper presents DIAL, the motivation is to achieve robustness of zero-shot classification of VLM like CLIP. For the disentanglement, the author proposes the use of a spare autoencoder to decompose the attribution in the column space of the decoder. For identification, the author proposes the attribution score such that it can align the feature vector with the spurious concept without using spurious labels. To remove the spurious feature, the author first finds the space spanned by the spurious vector, then projects the visual embedding to this space and removes such components. The author conducts benchmark dataset evaluation on the Group robustness and overall performance. The results show it surpasses the existing SOTA, like TIE or Orth-Cali.

**Strengths:**

I found this paper quite interesting.

**1** It combines the recent trend of explainability in LLM that use a sparse autoencoder to disentangle the feature vector, and align with a post-hoc explanation to find spurious feature direction.

**2** The paper is well presented, and the logical flow of the paper is good. The overall soundness of the paper is good for me.

**3** The author conducted the benchmark evaluation, which is at least comparable with the existing work.

**Weaknesses:**

1. In Figure 4, I don't quite understand why the average Acc can be even lower than WGA  in FMOW.

2. To my knowledge, sparse autoencoders (SAE) are often used in the explanation of the transformer encoder's FFN layer. I don't know how the performance would be that migrates the SAE to explain the latent representation.

3. Correct me if I am wrong. I found the SAE uses pre-trained weights. Would there be any distribution shift to align with the spurious vector in the CLIP models? Why don't we train on the specific dataset?

4. Equation at line 208, I think both parts could be sort of the attribution score. For the first term, it shows how the activation of the feature towards the positive spurious concept. For the second term, it shows how the feature vector aligns with the spurious text prompt. Then my question is, why do we need both terms multiplied together? Have you tried just using a single term in ablation?

5. In line 239, why do we use the reconstructed embedding to remove the spurious vector, not the original embedding?

**Questions:**

(1) Related to weakness 3, I would like to see the dataset-specific SAE on the outcomes.

(2) I don't quite get how you evaluate the explaniblity, as you mentioned this method is fully interpretable.

(3) Figure 2 is a good motivation figure. I would also want to see the heatmap after applying this method.

(4) I am also curious how the alignment between the spurious vector you found based on SAE and the spurious vector found by TIE using the spurious text prompt?

(5) Can the method find the novel spurious feature? Like in ISIC, there are multiple spurious features. Can we apply DIAL to find and mitigate such spurious features?

**Details Of Ethics Concerns:**

I don't have any ethics concerns.

---

> ### Author Response · Authors · 2025-11-20
> **Official Comment by Authors to Reviewer TSFp - 1/2**
>
> We thank reviewer TSFp for their feedback, recognizing the strengths of the paper. We have clarified the point raised on use of SAEs for latent representations, added analysis on using pre-trained vs dataset specific SAE, and answered the points raised. The paper is updated based on the comments as required.
>
> &nbsp;
>
> **W-1:**
>
> Thank you for pointing this out, it was due the result of typographical error while plotting the figures. The plots with correct values (4.9/2.0) are updated in the revised paper. This does not have any effect on that ablation experiment conclusion.
>
> &nbsp;
>
> **W-2:**
>
> To address this comment we would like to cite recent works which sucessfully used SAEs for interpreting latents from CLIP [1, 2, 3], ViTs, Dinov2, and SigLIP [4]. Additionally [4] have demonstrated the human alignment of the CLIP concepts disentangled through SAEs.
>
>
> [1] Zaigrajew, Vladimir, Hubert Baniecki, and Przemyslaw Biecek. "Interpreting CLIP with Hierarchical Sparse Autoencoders." Forty-second International Conference on Machine Learning.
>
> [2] Pach, Mateusz, et al. "Sparse autoencoders learn monosemantic features in vision-language models." arXiv preprint arXiv:2504.02821 (2025).
>
> [3] Lim, Hyesu, et al. "Sparse autoencoders reveal selective remapping of visual concepts during adaptation." The Thirteenth International Conference on Learning Representations.
>
> [4] Thasarathan, Harrish, et al. "Universal sparse autoencoders: Interpretable cross-model concept alignment." Forty-second International Conference on Machine Learning. 2025.
>
> &nbsp;
>
> **W-3 / Q-1:**
>
> As the pre-trained SAE we used on non medical datasets is trained CC3M a diverse and general dataset, it is highly likely to have the required concepts to reconstruct embeddings from CelebA, Waterbirds, FMOW and many other general datasets. Similarly for medical datasets we used an SAE trained on PMC-15M a large scale image-text pair dataset. We were able to find and mitigate the spurious features within the pre-trained SAE feature dictionary although training on specific dataset might yeild even better results if we have large enough training data.  The reasoning behind not training SAE on specific debiasing dataset initially is mainly the small dataset size which could limit the SAE quality.
>
> Following your comment, we trained SAEs from scratch on CelebA (140k training images), and waterbirds datasets (4.6k training images) and found them to improve the worst group accuracy for both these datasets. While the performance of on CelebA is close to the one where pre-trained SAE is used, waterbirds does little worse owing to small training set.
> This suggests that training SAE on the specific debiasing dataset and using it for spurious correlation mitigation especially when pre-trained SAE is not available is feasible. Below table shows the results on specific debiasing datasets.
>
> &nbsp;
>
> **Table: Evaluation of our framework when SAE is trained with debiasing dataset**
>
> | Dataset | Original (Avg/WG) ($\uparrow$) | DIAL (Avg/WG) ($\uparrow$) |
> | :--- | :---: | :---: |
> | CelebA | 81.20/73.35 | **86.41/84.71** |
> | Waterbirds | **83.72**/31.93 | 75.2/**49.31** |
> | FMOW | 26.05/18.16 | **26.67/18.16** |
>
> &nbsp;
>
> **W-4:**
>
> Motivated by prior work on identifying SAE features [1], we used both mean activation difference and cosine similarity. Prompted by your comment, we conducted ablation experiments which suggest that mean activation difference alone yields performance comparable to using both.
>
> &nbsp;
>
>
> **Table: Ablation study for identifying SAE features.**
>
> | Model - Dataset | Mean Activation Diff | Cosine Similarity | Both |
> | :--- | :---: | :---: | :---: |
> | CelebA - ViT L/14 | 86.86/**85.39** | 86.78/85.35 | **86.87**/85.24 |
> | CelebA - ViT B/32 | 85.54/83.33 | 85.28/82.78 | **85.54/83.47** |
> | Waterbirds - ViT L/14 | **82.61/68.85** | 74.95/50.29 | 82.6/68.69 |
> | Waterbirds - ViT B/32 | 71.85/52.68 | 70.69/48.12 | **71.88/52.82** |
> | ISIC | **70.71/68.42** | 62.45/55.84 | **70.71/68.42** |
> | Covid-19 | **61.11/48.28** | 58.33/34.48 | **61.11/48.28** |
>
> &nbsp;
>
>
> [1] Karvonen, Adam, et al. "Evaluating sparse autoencoders on targeted concept erasure tasks, 2024." URL https://arxiv. org/abs/2411.18895.
>
>
> &nbsp;
>
> **W-5:**
> As the reconstructed embedding is more aligned with the feature directions learned by SAE, we have decided to use the reconstructed embedding instead of the original embedding for the subspace removal. Also, this can help the framework to be robust when an low quality SAE with higher reconstruction loss is used
>
>
> &nbsp;

---

> ### Author Response · Authors · 2025-11-20
> **Official Comment by Authors to Reviewer TSFp - 2/2**
>
> **Q-2:**
>
> We mentioned the framework is fully interpretable because the spurious features, spurious subspace, and the entire removal mechanism can be examined and interpreted by a human as the indivudal spurious features are highly monosemantic. The interpretability of our framework stems from the highly monosemantic features learned by the SAE. Recent work [1] have evaluated the human interpretability of the features learned by SAEs using Monosemanticity Scores (MS score) ranging from 0-1, with higher scores being more interpretable to the humans. Based on experiments from [1] MSAE with CLIP ViT L/14 which we used for CelebA, Waterbirds, and FMOW has an MS score of 0.97.
>
>
> [1] Pach, Mateusz, et al. "Sparse autoencoders learn monosemantic features in vision-language models." arXiv preprint arXiv:2504.02821 (2025).
>
>
>
> &nbsp;
>
> **Q-3**
> We have added a figure in the paper  (Figure 5 in Appendix) which shows heatmap before and after applying the method.
>
> &nbsp;
>
> **Q-4:**
> We have calculated the alignment between spurious vector used by TIE and the spurious subspace found by our method.
> For CelebA dataset, the cosine similarity between the Male spurious feature of TIE and subspace is 0.34 while for Female it is 0.35.
>
>
> &nbsp;
>
>
> **Q-5:**
> With our new DIAL+ framework we can detect unknown spurious features in any dataset including ISIC. Currently we have not yet experimented with concept naming methods to name these detected spurious feature directions.

---

> > ### Comment · Reviewer_TSFp · 2025-11-26
> >
> > The reviewer thank you for your very detailed response. Overall, I am satisfied with your additional content. There is only one thing we should carefully discuss. I didn't quite get your response:
> >
> > ''We mentioned the framework is fully interpretable because the spurious features, spurious subspace, and the entire removal mechanism can be examined and interpreted by a human as the indivudal spurious features are highly monosemantic. The interpretability of our framework stems from the highly monosemantic features learned by the SAE. Recent work [1] have evaluated the human interpretability of the features learned by SAEs using Monosemanticity Scores (MS score) ranging from 0-1, with higher scores being more interpretable to the humans. Based on experiments from [1] MSAE with CLIP ViT L/14 which we used for CelebA, Waterbirds, and FMOW has an MS score of 0.97.''
> >
> > Why is the MS score a good metric for evaluating explainability? I mean, all the datasets you mentioned are pretty monosemantic, so they will definitely show high MS scores. We need to discuss the explainability of your framework systematically. I am open to raising my score if my full concerns are addressed. Thank you for your effort.

---

> ### Author Response · Authors · 2025-11-27
> **Reply by Authors to Reviewer TSFp Comments**
>
> We thank you for your comment and the opportunity to clarify our evaluation of interpretability.
>
> &nbsp;
>
> To address your concern regarding the dependency of the MS score on our specific debiasing datasets: We do not compute the MS score on the specific debiasing datasets (e.g., CelebA, Waterbirds, FMOW, ISIC, Covid). Instead, we compute the MS score on the validation sets of the SAEs (ImageNet validation set for the standard vision backbones and PMC-15M validation set for the medical backbone). These validation sets contain a much wider variety of concepts and much higher complexity than the  environments of our debiasing datasets.
>
> Therefore, the interpretability scores reported reflect the general quality of the features learned by the SAE, rather than an artifact of the debiasing datasets.
>
> In our previous response, we referenced the MS score (0.97 for ViT L/14) reported by [1] to illustrate the interpretability of SAEs.  They report the highest MS score obtained for a neuron in an SAE on Imagenet validation set.
>
> &nbsp;
>
> **Quantitative Evaluation:** For a through evaluation, we now report the specific average per-neuron MS scores calculated for the three SAEs used for our main results in our Tables 1-4 of the paper.
>
> We computed the MS scores using the automated method proposed in [1] across all neurons in our SAEs (We used ImageNet validation dataset for (B/32, and L/14) and PMC 15M validation dataset for BiomedCLIP).
>
> | SAE Backbone Model | MS Score (Avg) |
> | :--- | :---: |
> | CLIP ViT-B/32 | 0.56 |
> | CLIP ViT-L/14 | 0.43 |
> | BiomedCLIP | 0.42 |
>
> &nbsp;
>
> **Mapping Scores to Human Interpretability:** To contextualize these numbers, we refer to the user study conducted by [1], which calibrates MS scores against actual human perception. Their study establishes that an MS score in the range of 0.4–0.5 corresponds to a human alignment rate of approximately 65–70\%. These results indicate that the features learned by the SAEs we used are largely monosemantic and align with human perception. Consequently, this semantic coherence translates directly into explainability for our framework because the underlying units represent interpretable concepts, humans can explicitly interpret the spurious feature directions identified by our model. This also enables a verifiable workflow where users can inspect the the entire mitigation process.
>
> &nbsp;
>
> [1]Pach, M. et al. (2025) ‘Sparse Autoencoders Learn Monosemantic Features in Vision-Language Models’, in The Thirty-ninth Annual Conference on Neural Information Processing Systems. Available at: https://openreview.net/forum?id=DaNnkQJSQf.

---

### Official Review · Reviewer_bU24 · 2025-10-29

**Soundness:** 3
**Presentation:** 3
**Contribution:** 3
**Rating:** 6
**Confidence:** 5

**Summary:**

The paper presents a novel, zero-shot method named DIAL to mitigate spurious correlations in Vision-Language Models. DIAL uses the VLM's own zero-shot predictions to create pseudo-labels for identifying samples likely affected by a known spurious attribute and leverages a pre-trained Sparse Autoencoder to decompose VLM embeddings to gain more disentangled, interpretable features that correspond to the spurious attribute.
The authors validate DIAL on five benchmark datasets using multiple VLM backbones. The results show that DIAL consistently improves worst-group accuracy over baselines.

**Strengths:**

1. **Comprehensive Empirical Evaluation:** The experimental setup is thorough and convincing.
The use of five standard and diverse benchmark datasets, including challenging medical and real-world scenarios, demonstrates the method's broad applicability.

2. **Clarity and Presentation**: The paper is well-written, logically structured, and easy to follow.

**Weaknesses:**

1. **Dependency on Pre-trained SAEs**: The method's effectiveness is largely affected by the quality of a pre-trained SAE for the given VLM backbone. The paper does not discuss the sensitivity of DIAL to the SAE's quality (e.g., degree of disentanglement, reconstruction error, sparsity level). If a high-quality, pre-trained SAE is not available for a particular VLM, the contribution of DIAL is limited.  Plus, I'm quite curious about which kinds of SAE models[1,2,3] are most suitable for spurious tasks.

2. **Requirement of a Spurious Concept Description:** While label-free, the method still requires a user to provide a high-level textual description of the spurious attributes (e.g., "Male", "Female"). This assumes that the source of spurious correlation is known, which means that the method cannot discover unknown or hard-to-describe spurious features (e.g., a subtle imaging artifact without a common name). This limitation should be explicitly stated.

3. **Potential for Negative Interference:**  The orthogonal projection forcefully removes any information in the direction of the spurious subspace. If a genuinely causal feature is closely aligned with a spurious one in the embedding space, this process could inadvertently harm model performance by removing useful information. I believe the paper should give a more detailed discussion or analysis of this potential failure mode.

[1] BatchTopK SAE: Bussmann, Bart, Patrick Leask, and Neel Nanda. "Batchtopk sparse autoencoders." arXiv preprint arXiv:2412.06410 (2024).

[2] JumpReLU SAE: Rajamanoharan, Senthooran, et al. "Jumping ahead: Improving reconstruction fidelity with jumprelu sparse autoencoders." arXiv preprint arXiv:2407.14435 (2024).

[3] SAE + Contrastive loss: Wen, Tiansheng, et al. "Beyond matryoshka: Revisiting sparse coding for adaptive representation." arXiv preprint arXiv:2503.01776 (2025).

**Questions:**

The paper's utilization of SAE with KNN for identifying useful sparse representations for downstream tasks is a technique that has been explored in prior literature. For example, the following papers should be discussed in the related work:

[1] Tian, Zhihua, et al. "Sparse autoencoder as a zero-shot classifier for concept erasing in text-to-image diffusion models." arXiv preprint arXiv:2503.09446 (2025).

[2] Wen, Tiansheng, et al. "Beyond matryoshka: Revisiting sparse coding for adaptive representation." arXiv preprint arXiv:2503.01776 (2025).

---

> ### Author Response · Authors · 2025-11-20
> **Official Comment by Authors to Reviewer bU24 - 1/2**
>
> We thank reviewer bU24 for their review and recognizing the broad applicability of our method. To address the concerns raised we have added analysis w.r.t SAE quality, and possible ways circumvent the requirement of a pre-trained SAE. We have extended our framework to include DIAL+ to overcome the limitation of requiring spurious concept descriptions. These results are included in the revised paper.
>
> &nbsp;
>
> **W-1: Dependency on Pre-trained SAEs**
>
> i) Sensitivity to SAE Quality:
>
> We have evaluated the mitigation performance of our framework using SAEs with different reconstruction quality, sparsity, and decoder orthogonality. We find that disentanglement capability (measured here using decoder orthogonality) is high for SAEs even when reconstruction loss is relatively higher. We show that even with varying sparsity levels and reconstruction loss our framework remains effective. We find that waterbirds is more sensitive to reconstruction loss than CelebA dataset. We observe WG accuracy is improved for all the cases. From these results we could deduct that if an SAE could reconstruct well enough to preserve the discriminative features while maintaining the high disentanglement we can mitigate the spurious correlations.
>
> &nbsp;
>
> **Table: Analysis of performance of our method w.r.t SAE Quality on CelebA dataset**
>
> | Recon. Loss ($\downarrow$) | Sparsity ($\uparrow$) | Decoder Orthogonality ($\downarrow$) | Avg/WG Acc ($\uparrow$) |
> | :---: | :---: | :---: | :---: |
> | 0.44 | 0.72 | 0.0016 | **87.35**/84.44 |
> | 0.012 | 0.51 | 0.0014 | 86.41/**84.71** |
> | 0.019 | 0.79 | 0.0016 | 86.48/84.65 |
>
>
> &nbsp;
>
>
> **Table: Analysis of performance of our method w.r.t SAE Quality on Waterbirds dataset**
>
> | Recon. Loss ($\downarrow$) | Sparsity ($\uparrow$) | Decoder Orthogonality ($\downarrow$) | Avg/WG Acc ($\uparrow$) |
> | :---: | :---: | :---: | :---: |
> | 0.44 | 0.72 | 0.0016 | 71.98/41.51 |
> | 0.075 | 0.49 | 0.0013 | 75.2/49.31 |
> | 0.019 | 0.79 | 0.0016 | **81.98/68.51** |
>
>
> &nbsp;
>
>
> ii) Availability of a Quality Pre-trained SAE:
>
> We agree that having a high quality pre-trained SAE can give the most optimal results, but we show even when a pre-trained SAE is not available just by training an SAE on the debiasing dataset itself have shown to get decent results as shown in the below table. The lower accuracy on waterbirds could be attributed to small training set used for training SAE.
>
> &nbsp;
>
>
> **Table: Evaluation of our framework when SAE is trained with debiasing dataset**
>
> | Dataset | Original (Avg/WG) ($\uparrow$) | DIAL (Avg/WG) ($\uparrow$) |
> | :--- | :---: | :---: |
> | CelebA | 81.20/73.35 | **86.41/84.71** |
> | Waterbirds | **83.72**/31.93 | 75.2/**49.31** |
> | FMOW | 26.05/18.16 | **26.67/18.16** |
>
>
> &nbsp;
>
>
> iii) We find that MSAE on average performs the best overall for spurious feature mitigation, which can possibly be attributed to the better quality of MSAE on various SAE evaluation metrics. Below table shows the performance of our framework with BatchTopKSAE, and JumpReLUSAE.
>
>
> &nbsp;
>
> **Table: Evaluation of our framework using different SAEs. Average Accuracy/Worst-Group Accuracy are reported.**
>
> | Dataset | BatchTopKSAE | JumpReLUSAE | MSAE |
> | :--- | :---: | :---: | :---: |
> | CelebA | 87.35/84.44 | **87.74**/84.44 | 86.87/**85.24** |
> | Waterbirds | 71.98/41.51 | 77.80/56.01 | **82.60/68.69** |
> | ISIC | 69.57/58.73 | **72.16**/66.67 | 70.71/**68.42** |
> | Covid-19 | 46.52/20.69 | 58.33/34.48 | **61.11/48.28** |
>
>
> &nbsp;

---

> ### Author Response · Authors · 2025-11-20
> **Official Comment by Authors to Reviewer bU24 - 2/2**
>
> **W-2: Requirement of a Spurious Concept Description**
>
> To address this limitation and improve the utility in real world, we have introduced DIAL+, a variant that operates without a priori knowledge of spurious attributes. This method described in more detail in the revised paper utilize pseudo-labels to identify samples that are closer to the centroid of a conflicting class (the "other" class) than to their own class centroid. These samples often reside near the decision boundary and are highly correlated with spurious signals as shown by prior work. Once we obtain these samples we determine the spurious features based on the most commonly activated features within this subset. This method (DIAL+) is shown to give results comparable to to DIAL.
>
> &nbsp;
>
> **Table: Performance of DIAL compared with DIAL+**
>
> | Dataset (Model) | DIAL (Avg/WG) ($\uparrow$) | DIAL+ (Avg/WG) ($\uparrow$) |
> | :--- | :---: | :---: |
> | CelebA (ViT-B/32) | 85.54/83.47 | 85.28/83.42 |
> | CelebA (ViT-L/14) | 86.87/85.24 | 86.54/85.15 |
> | Waterbirds (ViT-B/32) | 71.88/52.82 | 68.48/42.26 |
> | Waterbirds (ViT-L/14) | 82.60/68.69 | 82.25/69.18 |
> | FMOW | 26.09/19.90 | 26.23/19.24 |
> | ISIC | 70.71/68.42 | 68.93/65.45 |
> | COVID-19 | 61.11/48.28 | 61.11/48.28 |
>
> &nbsp;
>
>
>
> **Q-1:**
>
> We have updated the related work to include the suggested references. We want to highlight that we use KNN for finding the samples which are disproportionally effected by the spurious correlations. For finding the spurious feature directions within the SAE feature dictionary we use the attribution method described in section 3.2 of the paper.

---

> > ### Comment · Reviewer_bU24 · 2025-11-27
> >
> > Thank you for the detailed updates and new experiments. I appreciate the additional comparison with other SAEs. I have decided to keep my score.

---

### Official Review · Reviewer_aziT · 2025-11-01

**Soundness:** 3
**Presentation:** 3
**Contribution:** 2
**Rating:** 6
**Confidence:** 3

**Summary:**

VLMs often rely on spurious correlations, which can affect downstream tasks. The authors introduce DIAL, a zero-shot method that finds and mitigates spurious correlations. DIAL operates by (1) filtering representations that might be disproportionately influenced by spurious features, (2) applying a sparse autoencoder to disentangle the representations and identify feature directions associated with spurious feature, and (3) removing the subspace spanned by the spurious directions from the representations. Results across several benchmarks demonstrate the utility of the method.

**Strengths:**

- This work addresses an important problem - finding and mitigating spurious correlations learned by vision-language models
- Results show performance improvements when compared to several baselines, suggesting utility of the approach. The method also works well across domains (i.e. general domain as well as medical domain).

**Weaknesses:**

- **Insufficient analysis:** Section 4.4 provides overall metrics across various datasets, but does not provide sufficient fine-grained analysis of results. Ablations are also limited. Ultimately, it is not clear to me *why* the method works better than baselines.
- **Need for attribute labels:** The proposed method requires a set of candidate spurious attributes, which may not always be known ahead of time and might limit utility of the method in real-world settings.
- **Choice of sparse autoencoder:** The authors consider one off-the-shelf pretrained sparse autoencoder for their analyses. How robust are the results to different choices of the autoencoder?

**Questions:**

Questions are listed above under weaknesses.

---

> ### Author Response · Authors · 2025-11-20
> **Official Comment by Authors to Reviewer aziT - 1/2**
>
> We thank the reviewer aziT for their feedback, and recognizing the importance of the problem and cross-domain utility. In response to the points raised we have included more detailed analysis on the results, and extended our framework to include DIAL+ which works comparable to DIAL without a need for spurious attribute candidates. Please find our response for each of the comment made below. The revised paper includes the new results and analysis.
>
> &nbsp;
>
> **W-1: Insufficient analysis**
>
> The improved performance can be attributed to below mechanisms:
>
> **1. Minimized Feature Interference via Disentanglement:**
>
> Standard baselines often operate in dense, polysemantic embedding spaces. In such spaces, removing a spurious feature vector frequently degrades causal features due to feature superposition. By using Sparse Autoencoder (SAE) latent space, we leverage a highly disentangled representation where feature vectors are nearly orthogonal. This orthogonality allows us to surgically remove spurious features with minimal impact on the semantic integrity of causal features.
>
> **2. Selective Intervention using candidate selection:**
>
> Even within the SAE latent space, perfect orthogonality is not always achieved. Blanket removal of features across all samples can inadvertently harm "clean" samples (those not relying on spurious correlations). We explicitly identify samples that are disproportionately affected by spurious features. We apply our removal intervention only to these identified samples.
> We have included a new ablation (below table) comparing our selective approach against a global application of the method.
>
> &nbsp;
>
> **Table: Ablation study with and without candidate selection**
>
> | Dataset - Model | Without candidate selection | With Candidate selection |
> | :--- | :---: | :---: |
> | CelebA - ViT L/14 | 86.43/84.82 | **86.54/85.15** |
> | CelebA - ViT B/32 | 84.85/81.67 | **85.28/83.42** |
> | Waterbirds - ViT L/14 | 81.60/51.56 | **82.25/69.18** |
> | Waterbirds - ViT B/32 | 70.45/50.31 | **71.88/52.82** |
> | FMOW - ViT L/14 | 26.04/19.55 | **26.09/19.90** |
>
> &nbsp;
>
> **3. Subspace removal instead of feature ablation:**
>
> Removing the entire spurious subspace, rather than simply ablating the corresponding feature activation to zero, aids in removing unidentified spurious features that are highly aligned with that subspace. This results in a more effective elimination of spurious features, as demonstrated in Figure 4 of the paper.
>
> **4. Feature selection through Attribution Mass:**
>
> Since different backbone embeddings and SAEs exhibit varying activation patterns, using a fixed Top-K approach to select feature directions corresponding to spurious features was shown to be less effective than selection through attribution mass, as shown in Figure 3 of the paper.
>
> &nbsp;
>
> **W-2: Need for attribute labels**
>
> **To address this limitation and improve the utility in real world, we have introduced DIAL+**, a variant that operates without a priori knowledge of spurious attributes. This method described in more detail in the revised paper utilize pseudo-labels to identify samples that are closer to the centroid of a conflicting class (the "other" class) than to their own class centroid. These samples often reside near the decision boundary and are highly correlated with spurious signals as shown by prior work [1]. Once we obtain these samples we determine the spurious features based on the most commonly activated features within this subset. This method (DIAL+) is shown to give results comparable to to DIAL.
>
> &nbsp;
>
> **Table: Performance of DIAL compared with DIAL+**
>
> | Dataset (Model) | DIAL (Avg/WG) ($\uparrow$) | DIAL+ (Avg/WG) ($\uparrow$) |
> | :--- | :---: | :---: |
> | CelebA (ViT-B/32) | 85.54/83.47 | 85.28/83.42 |
> | CelebA (ViT-L/14) | 86.87/85.24 | 86.54/85.15 |
> | Waterbirds (ViT-B/32) | 71.88/52.82 | 68.48/42.26 |
> | Waterbirds (ViT-L/14) | 82.60/68.69 | 82.25/69.18 |
> | FMOW | 26.09/19.90 | 26.23/19.24 |
> | ISIC | 70.71/68.42 | 68.93/65.45 |
> | COVID-19 | 61.11/48.28 | 61.11/48.28 |
>
> &nbsp;
>
> [1] Li, Weiwei, et al. "Let Samples Speak: Mitigating Spurious Correlation by Exploiting the Clusterness of Samples." Proceedings of the Computer Vision and Pattern Recognition Conference. 2025.

---

> ### Author Response · Authors · 2025-11-20
> **Official Comment by Authors to Reviewer aziT - 2/2**
>
> **W-3: Choice of sparse autoencoder**
>
> We show results using other SAEs including BatchTopKReLU [1], JumpReLU [2], compared with MSAE. Overall we observe that although MSAE performs the best attributing to its better SAE quality, our results show that even with other SAEs our framework improves over the ZS baseline. The lower performance of BatchTopKSAE for Covid dataset could be attributed to its higher reconstruction loss.
>
> &nbsp;
>
> **Table: Evaluation of our framework using different SAEs. Average Accuracy/Worst-Group Accuracy are reported.**
>
> | Dataset | BatchTopKSAE | JumpReLUSAE | MSAE |
> | :--- | :---: | :---: | :---: |
> | CelebA | 87.35/84.44 | **87.74**/84.44 | 86.87/**85.24** |
> | Waterbirds | 71.98/41.51 | 77.80/56.01 | **82.60/68.69** |
> | ISIC | 69.57/58.73 | **72.16**/66.67 | 70.71/**68.42** |
> | Covid-19 | 46.52/20.69 | 58.33/34.48 | **61.11/48.28** |
>
>
> &nbsp;
>
> We also evaluated our framework on training MSAE using the training set of specific debiasing dataset we are using for evaluation. This can be an alternative when off the shelf pre-trained SAEs are not available. The results are shown below. These results shows the robustness of our approach to SAEs trained with smaller and specific training sets. Lower performance on waterbird dataset could be owed to small training set of the waterbirds (around 4.6k).
>
> &nbsp;
>
> **Table: Evaluation of our framework when SAE is trained with debiasing dataset**
>
> | Dataset | Original (Avg/WG) ($\uparrow$) | DIAL (Avg/WG) ($\uparrow$) |
> | :--- | :---: | :---: |
> | CelebA | 81.20/73.35 | **86.41/84.71** |
> | Waterbirds | **83.72**/31.93 | 75.2/**49.31** |
> | FMOW | 26.05/18.16 | **26.67/18.16** |
>
> &nbsp;
>
> [1] BatchTopK SAE: Bussmann, Bart, Patrick Leask, and Neel Nanda. "Batchtopk sparse autoencoders." arXiv preprint arXiv:2412.06410 (2024).
>
> [2] JumpReLU SAE: Rajamanoharan, Senthooran, et al. "Jumping ahead: Improving reconstruction fidelity with jumprelu sparse autoencoders." arXiv preprint arXiv:2407.14435 (2024).

---

### Official Review · Reviewer_ddFq · 2025-11-01

**Soundness:** 2
**Presentation:** 2
**Contribution:** 2
**Rating:** 2
**Confidence:** 4

**Summary:**

This paper proposes DIAL, a label-free and zero-shot method to mitigate spurious correlations in vision-language models (VLMs). The approach uses sparse autoencoders to disentangle image embeddings and identify feature directions associated with spurious attributes. These directions are then removed via orthogonal projection to produce debiased representations. The method is evaluated on several benchmark datasets and compared against existing zero-shot debiasing techniques.

**Strengths:**

1. The method is fully zero-shot and does not require labeled data, retraining, or external models, which improves scalability.
2. The use of sparse autoencoders for disentangling representations is well-motivated and contributes to interpretability.

**Weaknesses:**

1. Limited novelty: The core idea—removing spurious directions via projection—is conceptually similar to prior work. The use of sparse autoencoders is incremental and not fundamentally new in the context of representation disentanglement.

2. Low practical impact: The spurious correlation issues addressed (e.g., background bias in Waterbirds, gender bias in CelebA) are well-known and have been extensively studied. The paper does not convincingly demonstrate that these issues remain critical in modern VLMs.

3. Outdated model focus: The analysis centers on older VLMs like CLIP ViT-B. It remains unclear whether the same spurious correlation problems persist in newer models such as SigLIP, OpenCLIP, or multi-modal transformers trained with more diverse data.

4. Assumption-heavy candidate selection: The method relies on pseudo-labels and centroid-based heuristics to identify biased samples, which may be unreliable in real-world settings or for more complex tasks.

5. Lack of generalization evidence: The paper does not explore whether the proposed mitigation transfers across tasks (e.g., retrieval, captioning) or domains beyond the selected benchmarks.

**Questions:**

Can your method be extended to mitigate spurious correlations in text embeddings or multi-modal fusion layers?

Is there any evidence that your projection-based debiasing improves downstream task performance beyond classification?

---

> ### Author Response · Authors · 2025-11-20
> **Official Comment by Authors to Reviewer ddFq - 1/2**
>
> We thank the reviewer ddFq for acknowledging the interpretability and scalability of our zero-shot approach, and we appreciate their critical feedback on evaluating on newer backbones, and other downstream tasks which we have addressed.  Please find our detailed response for each of the comment made below. The revised paper includes the new results.
>
>
> &nbsp;
>
> **W-1: Limited novelty**
>
> Although orthogonal projection is an established technique for removing directions, our novelty lies in the **label free, zero-shot discovery of these spurious directions using SAEs and effectively removing them within an VLM framework**. Specifically, we introduce:
>
> 1) A mechanism to select samples likely affected by bias in a zero-shot setting.
>
> 2) A method to identify specific spurious directions from SAE dictionary and effective removal of the spurious subspace.
>
> 3) **Update to original submission:** A lightweight extension of our framework to operate **without spurious attribute candidate sets by adding an automated detection capability**. Our updated results demonstrate that this fully automated version achieves performance close to our original method (DIAL), advancing the state-of-the-art in label free debiasing. With this addition, we also want to highlight that our **framework is the only one among the baselines which does zero-shot detection and mitigation of spurious correlations without any requirement of additional data, labels or use of LLMs.**
>
> &nbsp;
>
> **W-2: Low practical impact**
>
> 1) We want to highlight that in addition to Waterbirds and CelebA, **our framework demonstrates efficacy on challenging real-world domains, including Medical Imaging (ISIC Skin Cancer, Covid-19 Chest X-Rays) and Satellite Imagery (FMOW).** While models like BiomedCLIP [4] show promise for medical tasks, their tendency to learn spurious correlations undermines their reliability, making effective debiasing strategies essential for clinical adoption.
>
> 2) Prior work [1] has demonstrated that VLMs like CLIP regardless of the scale suffer from spurious correlations. There are works [2] which have given insights into how the spurious correlations can effect tasks where CLIP encoders are employed. And generally prior works [3] have shown the drawbacks of the CLIP vision encoder can transmit to multimodal LLMs.
>
>
> [1] Wang, Qizhou, et al. "A sober look at the robustness of clips to spurious features." Advances in Neural Information Processing Systems 37 (2024)
>
> [2] Zhou, Xinyang, et al. "The Devil is in the Spurious Correlations: Boosting Moment Retrieval with Dynamic Learning." arXiv preprint arXiv:2501.07305 (2025).
>
> [3] Tong, Shengbang, et al. "Eyes wide shut? exploring the visual shortcomings of multimodal llms." Proceedings of the IEEE/CVF Conference on Computer Vision and Pattern Recognition. 2024.
>
> [4] Zhang, Sheng, et al. "BiomedCLIP: A multimodal biomedical foundation model pretrained from fifteen million scientific image-text pairs. arXiv 2023." arXiv preprint arXiv:2303.00915 (2023).
>
> &nbsp;
>
> **W-3: Outdated model focus**
>
> **We note that our original backbones (which already includes OpenClip model ViT L-14 (LAION-2B) and BiomedCLIP) remain standard in the most recent baseline [1]**. However, based on this feedback, we have expanded our evaluation to include the best performing newer models from OpenCLIP.
>
> We have now evaluated DIAL on:
>
> SigLIP: ViT-SO400M-14-SigLIP-384 (trained on WebLI).
>
> EVA: EVA02-E-14-plus (trained on Laion 2B).
>
> ViT-H-14: ViT-H-14-quickgelu (trained on DFN5B).
>
>
> **Result:** We observe significant performance disparities across groups in these newer models as well. DIAL consistently improves the worst-group accuracy across CelebA, Waterbirds, and FMOW for these architectures.
>
> &nbsp;
>
> **Table 1: Evaluation of our framework with SigLIP (ViT-SO400M-14-SigLIP-384)**
>
> | Dataset | Original (Avg/WG) ($\uparrow$) | DIAL (Avg/WG) ($\uparrow$) |
> | :--- | :---: | :---: |
> | CelebA | 82.51/79.11 | **84.32/82.02** |
> | Waterbirds | **80.92**/61.37 | 80.54/**66.16** |
> | FMOW | **34.53**/25.18 | 34.10/**25.60** |
>
> &nbsp;
>
> **Table 2: Evaluation of our framework with EVA02-E-14-plus**
>
> | Dataset | Original (Avg/WG) ($\uparrow$) | DIAL (Avg/WG) ($\uparrow$) |
> | :--- | :---: | :---: |
> | CelebA | 84.78/80.54 | **87.76/86.52** |
> | Waterbirds | **76.95**/37.85 | 76.78/**56.65** |
> | FMOW | **29.62**/15.97 | 29.30/**16.70** |
>
> &nbsp;
>
> **Table 3: Evaluation of our framework with ViT-H-14-quickgelu**
>
> | Dataset | Original (Avg/WG) ($\uparrow$) | DIAL (Avg/WG) ($\uparrow$) |
> | :--- | :---: | :---: |
> | CelebA | 83.80/80.00 | **83.81/81.59** |
> | Waterbirds | 85.60/51.09 | **88.47/63.86** |
> | FMOW | 30.21/19.44 | **30.57/19.90** |
>
> &nbsp;
>
> [1] Lu, S., Chai, J., & Wang, X. (2025). Mitigating Spurious Correlations in Zero-Shot Multimodal Models. In The Thirteenth International Conference on Learning Representations.

---

> ### Author Response · Authors · 2025-11-20
> **Official Comment by Authors to Reviewer ddFq - 2/2**
>
> &nbsp;
>
> **W-4: Assumption-heavy candidate selection:**
>
> We acknowledge that using pseudo labels in settings where the model originally has low zero shot accuracy can be a noisy indicators of the biased samples.
>
> However, our method demonstrates the capability to work under these conditions. On the FMOW dataset, where the original model has a low zero-shot accuracy of 26.02%, our framework was shown to mitigate bias and improves performance compared to baselines.
>
> &nbsp;
>
> **W-5/Q-2: Lack of generalization evidence:**
>
> To address the concern regarding generalization and the reviewer's question about downstream tasks beyond classification, **we have extended our evaluation to image retrieval.**
>
> **Setup:** Following [1], we evaluated the debiasing capability of our framework using the MaxSkew@K metric [3] on the FairFace dataset [2].
>
> **Results:** We observe consistent improvements in embeddings debiased by DIAL w.r.t Age, Gender, and Ethnicity compared to the original zero-shot model (ViT L/14 - LAION 2B). This demonstrates the effectiveness of DIAL in improving the fairness in image retrieval tasks.
>
> &nbsp;
>
> **Table 4: Evaluation of our framework with retrieval task on FairFace.**
>
> | Sensitive Feature | Original (MaxSkew@1000) ($\downarrow$) | DIAL (MaxSkew@1000) ($\downarrow$) |
> | :--- | :---: | :---: |
> | Age | 1.32 | **0.95** |
> | Gender | 0.30 | **0.11** |
> | Ethnicity | 0.61 | **0.32** |
>
> &nbsp;
>
> [1] Chuang, Ching-Yao, et al. "Debiasing vision-language models via biased prompts." arXiv preprint arXiv:2302.00070 (2023).
>
> [2] Kimmo Kärkkäinen and Jungseock Joo. Fairface: Face attribute dataset for balanced race, gender, and age. arXiv preprint arXiv:1908.04913, 2019.
>
> [3] Sahin Cem Geyik, Stuart Ambler, and Krishnaram Kenthapadi. Fairness-aware ranking in search & recommendation systems with application to linkedin talent search. In Proceedings of the 25th acm sigkdd international conference on knowledge discovery & data mining, pages 2221–2231, 2019.
>
> &nbsp;
>
> **Q-1:**
>
> Yes, our method could be applied to debias fusion layer and text embeddings as our method does not make any assumptions on the layer or the modality of the embedding. To provide evidence for this claim, we performed an ablation where we debias the text embeddings instead of image embeddings using the same identified spurious feature directions as the standard method. Below are the results for this experiment comparing the original embeddings and the DIAL applied to text embeddings.
>
> &nbsp;
> **Table 5: Performance comparison of original embeddings with DIAL applied to text embeddings**
>
> | Dataset | Original (Avg/WG) ($\uparrow$) |         DIAL - text (Avg/WG) ($\uparrow$)|
> | :--- | :---: | :---: |
> | ISIC | **70.21**/42.21 | 66.50/**62.47** |
> | CelebA | 81.20/73.35 |**84.49/81.54** |

---

### Author Response · Authors · 2025-11-20
**General Comment by Authors to Reviewers**

We would like to thank all the reviewers for their constructive feedback and insights. We have added the responses to the concerns raised for each review and the paper is revised to include the corresponding changes. Some of the general updates are summarized below

&nbsp;


i) Introduction of DIAL+: Addressing the requirement of spurious feature candidate sets mentioned by multiple reviewers, we extended our framework to include DIAL+, a variant that detects spurious features without requiring prior spurious attribute information or descriptions. Our results show that DIAL+ achieves performance comparable to the original DIAL and other strong baselines.

&nbsp;

**Table: Performance of DIAL compared with DIAL+**

| Dataset (Model) | DIAL (Avg/WG) ($\uparrow$) | DIAL+ (Avg/WG) ($\uparrow$) |
| :--- | :---: | :---: |
| CelebA (ViT-B/32) | 85.54/83.47 | 85.28/83.42 |
| CelebA (ViT-L/14) | 86.87/85.24 | 86.54/85.15 |
| Waterbirds (ViT-B/32) | 71.88/52.82 | 68.48/42.26 |
| Waterbirds (ViT-L/14) | 82.60/68.69 | 82.25/69.18 |
| FMOW | 26.09/19.90 | 26.23/19.24 |
| ISIC | 70.71/68.42 | 68.93/65.45 |
| COVID-19 | 61.11/48.28 | 61.11/48.28 |

&nbsp;

ii) Expanded Evaluation: We validated our method on newer backbone architectures and extended the scope to image retrieval tasks, demonstrating consistent improvements over standard Zero-Shot (ZS) models.

&nbsp;

iii) Ablation Studies: We included additional experiments analyzing the impact of using different Sparse Autoencoders (SAEs) and evaluated the framework's utility under varying conditions.

---

### Author Response · Authors · 2025-12-03
**Summary Comment for Area Chair**

Through this comment, we provide an executive summary of our rebuttal and key revisions:

&nbsp;

**1. Introduction of DIAL+ (Addressing the Requirement for Spurious Attribute Candidates Concerns)**
* **Context:** Addressing feedback from Reviewers **aziT** and **bU24** regarding the reliance on pre-defined candidate sets for spurious attributes.
* **Update:** We developed and integrated **DIAL+**, a variant of our framework with spurious feature detection.
* **Result & Significance:** DIAL+ detects and mitigates spurious features without requiring prior spurious attribute candidates, achieving performance comparable to our original approach. Notably, this makes our framework the only approach among baselines to achieve zero-shot detection and mitigation without requiring additional data, labels, or use of LLMs.

&nbsp;

**2. Response to Novelty and Practical Impact Concerns (Reviewer ddFq)**
* **Novelty & Impact:** We have provided a detailed rebuttal to Reviewer **ddFq** regarding novelty and impact. We highlighted that the particular review point overlooked our effective validation on challenging **medical and satellite imaging** domains. We also added prior work references demonstrating the detrimental effect of spurious correlations on downstream VLM tasks to highlight the importance of the problem.
* **Backbones:** While the original backbones used for evaluation remain standard in current literature, we addressed concerns about models chosen by expanding our evaluation to **SigLIP, EVA-02, and ViT-H/14**.
* **Result:** Results confirm that these modern architectures still suffer from group disparities and that DIAL remains effective in mitigating them.

&nbsp;

**3. Generalization to Downstream Tasks**
* **Context:** Addressing Reviewer **ddFq**'s inquiry regarding utility beyond classification.
* **Update:** We extended our evaluation to **Image Retrieval** tasks.
* **Result:** Our debiased embeddings demonstrated significantly improved fairness metrics across demographic groups in retrieval scenarios, proving the cross-task generalization of our approach.

&nbsp;

**4. Expanded Ablation Studies and SAE Sensitivity**
* **Context:** Addressing reviewer feedback regarding the method's sensitivity to SAE quality, the choice of pre-trained SAEs, and the depth of the initial analysis.
* **Update:** We incorporated comprehensive experiments evaluating the impact of varying SAE qualities and architectures, alongside extended result analysis and ablation studies.
* **Result:** The expanded analysis confirms the robustness of the method across diverse SAE configurations.

&nbsp;

**5.** Reviewer **TSFp** expressed satisfaction with our initial responses and asked for further clarification on the framework's explainability; we addressed this by providing details on the metric and additional results.

---

### Meta-Review · Area_Chair_i1JK · 2026-01-07

**Summary:**

This paper presents DIAL, a zero-shot method to identify and mitigate spurious correlations in vision-language models (VLMs) using sparse autoencoders (SAEs). The approach disentangles image embeddings, isolates spurious feature directions, and removes them via orthogonal projection, all without requiring labeled data, model retraining, or external supervision. The authors have conducted a comprehensive evaluation across multiple standard and domain-specific benchmarks (including medical and satellite imagery), demonstrating consistent improvements in worst-group accuracy over existing zero-shot baselines. The authors have done comprehensive experiments in addressing the reviewers’ concerns. Given the rebuttal and the overall quality of this submission, the paper is above the acceptance threshold for ICLR.

**Reviewer Concerns:**

The initial reviews raised concerns about novelty, practical impact, reliance on older VLMs, assumption-heavy candidate selection, generalization to downstream tasks, and sensitivity to SAE quality. The authors provided substantial revisions and additional experiments, which effectively addressed the majority of the reviewers’ concerns:

Novelty & Extensions. The authors clarified that while projection-based debiasing is established, the core novelty lies in the zero-shot discovery of spurious directions via SAEs and a candidate selection mechanism. They further introduced DIAL+, a variant that automatically detects spurious attributes without predefined candidates, strengthening the claim of being a fully zero-shot and label-free framework.

Practical Impact & Model Relevance. The paper was extended to include newer VLM backbones (SigLIP, EVA, ViT-H), showing that spurious correlations persist even in recent large-scale models. The method was also validated on real-world domains (medical imaging, satellite imagery), underscoring its relevance for high-stakes applications.

Generalization & Downstream Utility. Additional experiments on image retrieval (FairFace dataset) demonstrate that DIAL improves fairness metrics beyond classification, addressing the question of broader applicability.

Technical Soundness. The authors provided detailed ablations justifying design choices (e.g., subspace removal vs. feature ablation, attribution scoring, selective intervention), and analyzed the impact of SAE quality, reconstruction loss, and decoder orthogonality. Experiments with different SAE architectures (BatchTopK, JumpReLU) and dataset-specific SAEs confirm the robustness of the approach.

Interpretability. The authors quantified interpretability via monosemanticity scores (MS) and referenced human-alignment studies, clarifying that the SAE features are genuinely interpretable and not an artifact of the evaluation datasets.

**Reviewer Scores:**

One reviewer gave a negative score, but the questions he/she raised were almost all verified by the authors' experiments.
Two reviewers gave positive ratings, while the last reviewer indicated that if the author solved the raised problems, he/she would be willing to give a higher rating.

---

### Decision · Program_Chairs · 2026-01-26

Accept (Poster)